# Insight into the Potential Mechanisms of Endocrine Disruption by Dietary Phytoestrogens in the Context of the Etiopathogenesis of Endometriosis

**DOI:** 10.3390/ijms241512195

**Published:** 2023-07-30

**Authors:** Dariusz Szukiewicz

**Affiliations:** Department of Biophysics, Physiology & Pathophysiology, Faculty of Health Sciences, Medical University of Warsaw, 02-004 Warsaw, Poland; dariusz.szukiewicz@wum.edu.pl

**Keywords:** endocrine disruption, phytoestrogens, endometriosis, endocrine-disrupting chemicals, etiopathogenesis of endometriosis, ectopic endometrium, dietary phytoestrogen intake, epigenetic factors

## Abstract

Phytoestrogens (PEs) are estrogen-like nonsteroidal compounds derived from plants (e.g., nuts, seeds, fruits, and vegetables) and fungi that are structurally similar to 17β-estradiol. PEs bind to all types of estrogen receptors, including ERα and ERβ receptors, nuclear receptors, and a membrane-bound estrogen receptor known as the G protein-coupled estrogen receptor (GPER). As endocrine-disrupting chemicals (EDCs) with pro- or antiestrogenic properties, PEs can potentially disrupt the hormonal regulation of homeostasis, resulting in developmental and reproductive abnormalities. However, a lack of PEs in the diet does not result in the development of deficiency symptoms. To properly assess the benefits and risks associated with the use of a PE-rich diet, it is necessary to distinguish between endocrine disruption (endocrine-mediated adverse effects) and nonspecific effects on the endocrine system. Endometriosis is an estrogen-dependent disease of unknown etiopathogenesis, in which tissue similar to the lining of the uterus (the endometrium) grows outside of the uterus with subsequent complications being manifested as a result of local inflammatory reactions. Endometriosis affects 10–15% of women of reproductive age and is associated with chronic pelvic pain, dysmenorrhea, dyspareunia, and infertility. In this review, the endocrine-disruptive actions of PEs are reviewed in the context of endometriosis to determine whether a PE-rich diet has a positive or negative effect on the risk and course of endometriosis.

## 1. Endocrine-Disrupting Chemicals (EDCs)

The endocrine system, in association with the nervous system and the immune system, regulates the body’s internal activities and interactions with the external environment to preserve the homeostasis of the internal environment [1,2]. Hormone-producing cells (both within endocrine glands and forming the disseminated endocrine system) secrete hormones (chemical messengers) that interact with specific targets (receptors), including those targets that are subjected to epigenetic modifications [2,3,4]. These interactions result in the regulation of a vast spectrum of functions, including the development, growth, energy balance (metabolism), reproduction, and regulation of body weight [3,4].

Organic compounds that (to varying degrees) resist photolytic, biological, and chemical degradation are called persistent organic pollutants (POPs) [5]. POPs are often halogenated and characterized by low water solubility and high lipid solubility, thus leading to their bioaccumulation in fatty tissues [5,6]. Due to the semivolatility of POPs and the physico-chemical characteristics that permit these compounds to occur either in the vapor phase or be adsorbed on atmospheric particles, the long-range transport of POPs through the atmosphere may be facilitated. Thus, POPs are ubiquitous throughout the world, even in regions where they have never been used [7]. The most commonly encountered POPs are organochlorine pesticides, such as DDT, industrial chemicals, polychlorinated biphenyls (PCBs), and unintentional byproducts of many industrial processes, especially polychlorinated dibenzo-p-dioxins (PCDDs) and dibenzofurans (PFDFs), which are commonly known as dioxins [8,9].

Many POPs are well known to interact with the endocrine system by mimicking, hindering, blocking, and promoting the normal activity of hormones [8,9,10,11]. Thus, these endocrine-disrupting chemicals (EDCs) are compounds in the environment (air, soil, or source of water), food, personal care products, and manufactured products that possess the ability to interfere with the normal function of the endocrine system [12,13]. EDCs may interfere with the synthesis, secretion, transport, binding, action, and metabolism of virtually all natural hormones in the body, including sex steroid hormones that correspondingly cause developmental and fertility problems, infertility, and hormone-sensitive cancers in women and men [13,14,15,16]. Specifically, exposure to EDCs above the threshold dose causes carcinogenic, neurotoxic, hepatotoxic, nephrotoxic, and immunotoxic effects, as well as teratogenic hazards with birth defects [17,18,19,20,21,22,23].

According to the Endocrine Society statement, endocrine disruptors can be defined as “an exogenous chemical, or mixture of chemicals, that can interfere with any aspect of hormone action” [24,25]. However, it is necessary to distinguish between endocrine disruption (endocrine-mediated adverse effects) and nonspecific effects on the endocrine system [26]. Endocrine disruption occurs as a consequence of the interaction of a chemical (classified as an EDC) with a specific molecular component of the endocrine system (for example, an estrogen receptor). In contrast, nonspecific effects on the endocrine system may be observed when systemic toxicity has a significant impact on homeostasis and indirectly perturbs endocrine signaling. When considering the integral nature of signaling pathways in the endocrine system, it is difficult to confidently distinguish endocrine disruption from transient fluctuations, adaptive/compensatory responses, or adverse effects on the endocrine system caused by mechanisms outside of the endocrine system that use nonendocrine-mediated modes of action [26,27]. This situation is further complicated by the fact that some organs/tissues can be affected by both endocrine and nonendocrine erroneous/disrupting signals.

Given that EDCs originate from many different sources, people may be exposed in many ways, including the air that they breathe, the food that they eat, and the water that they drink [25,28,29,30]. In addition, EDCs can enter the body via intact skin and mucous membranes [31]. Dietary intake is the main entry route of POPs and other EDCs into the human body and accounts for more than 90% of total chemical exposure [28,32]. Moreover, there is an increasing concern that permanent low-level exposure to EDCs may have adverse health impacts, particularly during fetal, neonatal, and childhood development. Therefore, important human health hazards should be expected in relation to EDCs, especially in the event of increasing environmental pollution [33,34,35,36]. Furthermore, it has been demonstrated that in addition to EDC, estrogen is a persistent compound in the environment. Estrogen contamination was confirmed in both lake water used for drinking and sewage water used for irrigation at concentrations that could affect plant growth (e.g., alfalfa) and sexual differentiation in fish [37,38,39]. These findings of estrogen as an environmental pollutant have been repeated and confirmed throughout the world, thus indicating that sex hormones, including estrogen and testosterone, are present in several environmental compartments, including soil and groundwater [40,41,42].

Chemicals with hormonal activity that may induce endocrine disruption can be divided into three main groups: synthetic compounds used in industry, agriculture, and consumer products; synthetic compounds used in the pharmaceutical industry (i.e., drugs); and natural compounds present in the food chain that contain phytoestrogens (PEs), i.e., compounds showing structural similarity to estradiol (E_2_) [14].

It should be clearly emphasized that, in this review, only the endocrine-disruptive actions of PEs will be reviewed in the context of endometriosis, which is an estrogen-dependent disease with still unknown etiology (see Section 2.2). General considerations on the effects of PEs as endocrine disruptors and estrogen-mediated alterations in endometriosis are followed by the current data on the role of orally administered PEs in the etiopathogenesis and course of endometriosis.

### 1.1. Phytoestrogens (PEs)

PEs, also called “dietary estrogens”, are estrogen-like nonsteroidal compounds derived from plants (e.g., nuts, seeds, fruits, and vegetables) and fungi, which are structurally similar to 17β-estradiol [43,44]. The estrogenic activity of PEs was first demonstrated in 1926; however, for the next 20 years, until fertility problems in sheep on isoflavone-rich diets were reported in Western Australia, it was uncertain as to whether they could have any effect on human or animal metabolism [44,45,46].

Based on the chemical structure, six main classes of PEs can be distinguished: flavonoids, stilbenes, enterolignans, coumestans, pterocarpans, and mycotoxins [47] (Table 1). Over 5000 naturally occurring flavonoids have been characterized from various plants. The main PEs derived from the diet are genistein, daidzein, and glycitein, which belong to a subclass of flavonoids called isoflavones [48]. PEs do not participate in any essential biological processes, and a lack of PEs in the diet does not result in the development of deficiency symptoms. Therefore, PEs are not considered nutrients [49].

In humans, after consuming PEs, they are converted in the gastrointestinal tract by complex enzymatic processes to heterocyclic phenols that are structurally similar to E2 [44]. Subsequently, absorbed phytoestrogen metabolites enter into enterohepatic circulation and may be excreted in the bile deconjugated by intestinal flora, reabsorbed, reconjugated by the liver, and excreted in the urine [44,50,51]. The concentrations of different phytoestrogen metabolites can vary widely between individuals, even when a controlled quantity of an isoflavone or lignan supplement is administered.

The structural similarity of PEs to endogenous estradiol E2 implies the presence of a phenolic ring that enables binding to estrogen receptors in humans. Other key structural elements that increase affinity for estrogen receptors and enable estrogen-like effects include low molecular weights similar to estrogens/E2 (MW = 272), optimal hydroxylation patterns, and (in the case of isoflavones) similarities of the E2 distances between two hydroxyl groups at the nucleus [52,53,54]. Analogous to estradiol, PEs bind to all known types of estrogen receptors, including ERα (NR3A1) and ERβ (NR3A2) receptors (which are the members of the superfamily class of nuclear receptors located in either the cell cytoplasm or nucleus) and a membrane-bound estrogen receptor known as G protein-coupled estrogen receptor (GPER), which is also known as G protein-coupled receptor 30 (GPR30) [55,56,57,58].

#### 1.1.1. Signaling via Nuclear Receptors

In humans, ERα is encoded by the gene *ESR1*, which is located on chromosome 6, locus 6q25.1, whereas ERβ is encoded by the *ESR2* gene located on chromosome 14 (14q23–24) [59,60]. In addition to the full-length isoforms, several shorter isoforms of ERs have been identified as a result of the presence of alternate start codons or as products of alternative splicing. The six crucial structural and functional domains of both ERα and ERβ were distinguished within the N-terminus (NTD: A/B domains, AF-1), DNA-binding domain (DBD or C domain), hinge (D domain), and C-terminal region containing the ligand-binding domain (LBD: E/F domain, AF-2) (Figure 1) [61,62,63].

The main, well-documented signaling pathways of estrogens are shown in a simplified manner in Figure 2.

Classical ligand-dependent ER activation results in the regulation of gene transcription in the nucleus or the activation of kinases in the cytoplasm (Figure 2A). This form of signaling mediates long-term genomic effects in estrogen-responsive tissues, including the human endometrium [59].

Estrogen binding to ERα or ERβ leads to the removal of the polyprotein inhibitory complex from the LBD with the release of heat shock protein 90 (Hsp90) and the induction of a conformational change, resulting in the homodimerization of the receptor. Crystallographic studies have shown that, in contrast to the classical binding characteristics of a substrate to its active site in an enzyme, the ligand-binding domain of the ERs is larger than the E_2_ molecule, which explains why it can accommodate a range of different-sized molecules, including those corresponding to PEs. Afterward, this signaling complex is translocated from the cytoplasm to the nucleus, where, after the recruitment of other coregulators, ERs act as ligand-activated transcription factors [60,61]. This direct genomic activity is associated with the binding of the DBD to the estrogen response element (ERE) on the target gene and the subsequent cis-activation of the enhancer of the target gene regulatory region that promotes transcription. In the “tethered” signaling pathway, ligand-activated ERs interact with other transcription factor (TF) complexes and attach to these transcription factors, which enables the indirect binding of the DBD to the ERE as TF-ERE [61,62]. The transcriptional activities of ERs are mediated by the coordinated action of their two activation domains, including the constitutive activation domain AF-1 at the N-terminus and the hormone-dependent AF-2 at the LBD. ERs have more than 30 synergistic activation factors, many of which are shared by nuclear receptors. The indirect regulation of gene transcription via the activation of the extracellular signal-regulated kinase 1/2 (ERK1/2) cascade and the phosphatidylinositol 3′-kinase (PI3K) signaling pathways is also involved in ERα/ERβ signaling [61,62,63].

The estrogenic effects of PEs are primarily mediated via ERα and ERβ (with higher affinity for Erβ) and by acting as agonists, partial agonists, and antagonists [69]. For example, isoflavone affinity for ERβ isoforms is approximately five times higher than the affinity for ERα isoforms, in contrast to E_2_, in which the affinities to both receptor types are generally the same [70,71,72].

Interesting results regarding phytoestrogen affinity for Erβ have originated by using molecular docking, which is a method that is frequently used in the process of computer-aided drug design (CADD) as a tool for the identification of novel and potent ligands, as well as for predicting the binding mode of already known ligands and for the comparative estimation/prediction of binding affinity [73]. In molecular docking, the most important aspect is the calculation of binding energy to fit a ligand in a binding site [74]. Comparisons between two or three complexes using the predicted binding energies as a criterion are commonly found in the literature [75,76,77]. Such studies have demonstrated that almost all popular herbal supplements contain phytochemical components that may bind to the human estrogen receptor and exhibit selective estrogen receptor modulation. For example, of the flavonoids, luteolin-8-propenoic acid has been shown to exhibit the strongest docking (most exothermic docking energies) to Erα, with a docking energy of −113.127 kJ/mol, which is more exothermic than those of E_2_, isoflavonoid genistein or mycotoxin zearalenone [78]. A common docking orientation for phenolic ligands in Erα is the hydrophobic acceptor pocket of Leu 387, Phe 404, Met 388, and Leu 391, along with edge-to-face π–π interactions with Phe 404 and hydrogen bonds between the phenolic –OH group and the guanidine group of Arg 394, as well as the carboxylate of Glu 356. The 7-OH group of this ligand can form an additional hydrogen bond with the carbonyl oxygen of Gly 521. No other flavonoid ligands showed notably strong docking with ERα [75]. However, without questioning the concept of phytoestrogen binding to ERs, some authors are concerned about the unreliability of binding energy comparisons between pairs of molecules using docking [79].

It has been proposed that the estrogenic or antiestrogenic activity of PEs may be determined by an individual’s amount of circulating endogenous estrogens, as well as the amount of bioavailable PEs and the number and type of ERs [80,81,82]. The approximately 100-fold lower affinity of PEs for ERs compared to human estrogens may be compensated for by their potentially high concentrations. For phytoestrogen levels that are several times higher than the concentration of endogenous estrogens, this higher affinity for ERβ may be even stronger than that exhibited by steroidal estrogens, which additionally suggests that PEs may exert their actions through distinctly different pathways [83,84]. The broad spectrum of estrogenic/antiestrogenic activity of PEs is due to the obvious fact that ERs have different functions. For example, ERα acts in cell proliferation, including carcinogenesis, whereas ERβ is responsible for cell cycle arrest, the modulation of the expression of many ERα-regulated genes, and the induction of multiple anticancer activities (e.g., apoptosis) [85,86,87,88]. Interestingly, some PEs have also demonstrated progesterone receptor activity [89].

ERα is predominantly expressed in the endometrium, breast cancer cells, ovarian stroma cells, efferent duct epithelium, and hypothalamus, whereas ERβ is expressed in the kidney, brain, bone, heart, lungs, intestinal mucosa, prostate, and endothelial cells [90,91,92]. Consequently, the preference of binding to ERα or ERβ by a given phytoestrogen may determine its tissue-selective biological effects, including endocrine disruption. Once bound, PEs exhibit selective ER modulator (SERMS) activity with a broad range of varying agonist/antagonist activities. The tissue-selective or tissue-specific effects depend significantly on the content and proportion of transcriptional coregulators (both coactivator and corepressor proteins) within the single cell. This indicates that in the case of predomination of coactivators in certain tissues, a given ligand may be an agonist of ERs, whereas a predominance of corepressors in another tissue releases the antagonistic effects of the same ligand [62]. Unlike the function of a cofactor to an enzyme, coregulators act as bridging or helper molecules that aid in forming large protein complexes to modulate appropriate activity on target gene chromatin. The detection of more than 200 coregulators for ER that are differentially expressed in many tissues can further confirm the tissue specificity of estrogen signaling [93]. Moreover, specific and unique conformational changes in the tertiary structure of ER inherently resulting from phytoestrogen binding can modulate the recruitment of coregulator proteins. Both coactivators and corepressors are crucial for the subsequent transcriptional activity of ER after its dimerization and binding to specific response elements known as estrogen response elements (EREs), which are present in the promotor region of target genes [62,93]. For example, genistein acting on ERβ is more efficient in enhancing the transcriptional activity of ERs compared to the stimulation of ERα. The observed difference is derived from the more efficient recruitment of the p160 (SRC) steroid receptor coactivators, TIF2 (SRC-2) and SRC-1a (NCoA-1), during ERβ activation. In general, the activation of ERβ has been shown to antagonize the cell-growth-promoting effects of ERα. This scenario may be of importance in highly estrogen-sensitive tissues, especially in ERα-overexpressing cancers (e.g., breast tumors), wherein a potential protective action against estrogen-dependent cancer remains closely related to the ratio of active ERβ versus ERα [93,94,95]. PEs bound to ERs can also activate transcription at AP-1 binding sites that bind Jun/Fos transcription factors [96].

#### 1.1.2. GPER Signaling

The classic perception of ER receptors as ligand-activated transcription factors mediating long-term genomic effects in hormonally regulated tissues has changed due to the fact that estrogens and PEs can also mediate rapid, nongenomic actions [97,98]. Such observations that the exposure of target tissue cells (including human endometrium) to estrogenic ligands can rapidly induce ion flows and the activation of various protein kinases across the plasma membrane independent of protein synthesis have led to the emergence of the concept of membrane ER [99]. Membrane-associated ER signaling pathways are typically associated with growth factor receptors and G protein-coupled receptors (GPCRs) [100,101]. A seven-transmembrane-domain receptor GPER (GPER1, first referred to as GPR30), which is a member of the G protein-coupled receptor (GPCR) superfamily, is one such first identified receptor that mediates estrogen-dependent kinase activation, as well as transcriptional responses [102,103]. Signaling through GPER occurs via the transactivation of the epidermal growth factor receptor (EGFR) and involves nonreceptor tyrosine kinases of the Src family [104]. The stimulation of GPER activates metalloproteinases and induces the release of heparin-binding epidermal growth factor-like growth factor (HB-EGF), which binds and activates EGFR, thus leading to the downstream activation of signaling molecules, such as the mitogen-activated protein kinases, ERK1 and ERK2 [104,105,106]. In addition, 17β-estradiol-mediated activation of GPER stimulates cAMP production, intracellular calcium mobilization, and PI3K activation [107,108]. The activation of signaling mechanisms involving cAMP, ERK, and PI3K may be responsible for the indirect transcriptional activity of GPER, which represents another regulatory function in addition to the abovementioned rapid signaling events [97]. The indirect nongenomic signaling pathway via membrane-associated GPERs with the transactivation of EGFRs is shown in Figure 2B.

Several PEs, including flavones (e.g., quercetin), isoflavones (e.g., genistein), lignans, coumestans, saponins, and stilbenes, can activate GPCRs [109]. For example, genistein and quercetin are able to stimulate c-fos expression in an ER-independent manner via GPER in ERβ-positive MCF7 and ERα-negative SKBR3 breast cancer cells [110]. However, PEs and mycoestrogens (e.g., zearalenone), even when displaying relatively high binding affinities for GPER and acting as agonists to increase cAMP synthesis, are more potent in activating ERα and ERβ [57,109]. In addition, some researchers have even suggested that the results obtained in vitro are not transferable to in vivo conditions; therefore, there is still a lack of evidence that GPER plays a significant role in mediating endogenous estrogen action in vivo [111]. The latter scenario may be due to the specificity of signaling via GPER and its intracellular localization. Namely, GPER is predominantly expressed on the membrane of the endoplasmic reticulum; thus, ligands must cross the plasma membrane to bind the receptor [112]. Thus, several studies have provided evidence demonstrating that a larger fraction of total cellular GPER is localized in intracellular compartments. These discrepancies regarding receptor localization may be partially caused by receptor trafficking between the endoplasmic reticulum and the plasma membrane during receptor biogenesis. The internalization of GPER in response to agonist stimulation should also be considered [113]. Moreover, GPER is made up of the same protein products of the genes that encode nuclear ERs. Specifically, membrane and nuclear ERs are derived from the same transcripts, but the former type is directed to the membrane via palmitoylation. The palmitoylation of the Cys447 residue of the ERα–ligand-binding domain (ERα-LBD) and Cys399 residue of ERβ-LBD through intermediary heat shock protein 27 enables the interaction of ERs with the caveolin-1 protein, which is required for the transport of GPER components to caveolae rafts within the cell membrane [63]. Palmitoylated ERs are translocated to the membrane as monomers, and the dimerization of GPER occurs within seconds of E_2_ exposure, which results in the activation of G protein α and βγ subunits (Gα and Gβγ, respectively) in a cell-type-dependent manner [63,114]. Subsequently, the depalmitoylation or weakening of the caveolin-1–receptor interaction causes the redistribution of ERs and their association with adaptors and/or signaling proteins, including proline-, glutamic-acid-, and leucine-rich protein 1 (PELP1), which is also known as a modulator of the non-genomic activity of ER (MNAR), proto-oncogene tyrosine-protein kinase Src, and tyrosine kinase receptors [63]. This correspondingly and, ultimately, contributes to the activation of the extracellular signal-regulated kinase/mitogen-activated protein kinase (ERK/MAPK) and phosphatidylinositol 3-kinase/serine–threonine kinase/mammalian target of rapamycin (PI3K/AKT/mTOR) signaling cascades, with respective effects on cellular proliferation, migration, and other estrogen-dependent processes [115,116].

#### 1.1.3. Signaling Not Mediated by ERs—A Significant Source of Differences in Bioactivity between E_2_ and PEs

Estrogens, including PEs, may also exert biological effects without interacting with ERs. The activation of ERs by ligand-independent mechanisms involves the recruitment of different sets of cofactors. In the ligand-independent signaling pathway, ERs are phosphorylated/activated by other active signaling cascades in a cell [66]. For example, growth factors or cyclic adenosine monophosphate (cAMP) activate receptor tyrosine kinases and intracellular kinase pathways, thus leading to MAPK activation with subsequent estrogen-independent phosphorylation of ERs (Figure 2C). This activation results in both direct ERE- and non-ERE-dependent genomic actions [67,68].

The activation of serotoninergic receptors and insulin-like growth factor receptor 1 (IGFR1), as well as the stimulation of free radical species binding and DNA methylation, are well-documented actions of PEs that do not involve ERs [46,55]. Moreover, in this mode of action of PEs, modified activities of tyrosine kinases, cycle adenosine monophosphate (cAMP), phosphatidylinositol-3 kinase/Akt, and mitogen-activated protein (MAP) kinase transcription of nuclear factor-kappa β (NF-κB) should be expected. Together with the confirmed participation of PEs in the regulation of the cell cycle and apoptosis via ERs, these ER-independent activities cause PEs to possess antioxidant, antiproliferative, antimutagenic, and antiangiogenic properties [117,118]. In clinical practice, this scenario translates into better or worse documented potential health benefits, including the alleviation of menopausal symptoms (e.g., hot flashes, night sweats, sleep problems, and mood changes) and a reduced risk of osteoporosis, heart disease, neurodegenerative processes, and breast cancer [46,119,120,121,122]. This last effect is still somewhat controversial because some clinical studies have reported data that suggest that isoflavones may increase breast cancer incidence in sensitive individuals via their estrogenic and proliferative effects [123,124,125]. The use of PEs in the prevention and management of type 2 diabetes is also the subject of clinical research [126].

### 1.2. Phytoestrogens (PEs) as Endocrine-Disrupting Chemicals (EDCs)

Adverse health effects should be expected following dietary intake of considerably high amounts of PEs because PEs may act as endocrine disruptors [127,128,129,130]. Consequently, the question of whether PEs are beneficial or harmful to human health remains unresolved. Given that the worldwide consumption of PEs is continually expanding, clarity on this subject is essential. The answer is likely complex and may depend on parameters such as age, health status, and even the presence or absence of specific gut microflora [130,131].

Similar to other EDCs, PEs exhibit a wide spectrum of abilities for disrupting the hormonal regulation of homeostasis. The most fundamental mechanisms of such potentially detrimental activity include the following: ❶ acting as a ligand at the binding sites of the hormone and mimicking the effects of the most specific endogenous ligand; ❷ antagonizing the effects of endogenous hormone by blocking its interaction at physiological binding sites; ❸ reacting directly and indirectly with a given hormone; ❹ altering the natural patterns of production and degradation of hormones; and ❺ disturbing cellular hormone receptor expression [132,133].

PEs behave as weak estrogen mimics or as antiestrogens. Despite the beneficial actions mentioned in the previous section, the supporting evidence that dietary intake of PEs is beneficial is indirect and inconsistent [47,48]. Moreover, it has been demonstrated that lifetime exposure to estrogen-like compounds, particularly during critical periods of development, has been associated with the formation of malignancies and several anomalies of the reproductive system [48]. PEs in maternal blood can pass through the placenta to the fetus in high amounts and can exert long-term effects, including adverse effects with consequences observed in postnatal life [134]. In addition, PEs are commonly found in pregnant women’s amniotic fluid. There is a sex difference in the concentrations, with higher levels observed in amniotic fluid containing female fetuses. This difference was not present in the maternal serum [135]. Moreover, soy ingestion increases amniotic fluid phytoestrogen concentrations in female and male fetuses [135]. The rapid transfer from the mother to the fetus was demonstrated for the phytoestrogen daidzein (which is an important representative of isoflavonoids in soya products) in pregnant rats. After the intravenous administration of daidzein to the mother, its concentration in the placental tissue and fetal liver amounted to 1/10 and 1/30 of the peak concentration of the maternal liver, respectively [134]. Exposure to a phytoestrogen-rich mesquite (*Prosopis* sp.) pod extract during the periconception and pregnancy periods in rats significantly affected the reproductive functions of male and female descendants. Furthermore, alterations in estrous cycles, decreased sexual behavior, estradiol and progesterone levels, and increased uterine and vaginal epithelia were observed in females. In males, a decrease in sexual behavior, testosterone, and sperm quality, as well as increased apoptosis in testicular cells, were reported [134]. All of these effects were similar to those caused by daidzein. These results may indicate that prenatal exposure to mesquite pod extract or daidzein administered to females before and during pregnancy can disrupt normal organization, activation, and behavioral programming with respect to reproductive physiology in female and male descendants [136,137].

The Ingestion of genistein, which is a soybean-originated isoflavone, may modulate leptin hormone, C-reactive protein, tyrosine kinase activities, and thyroid functions [138]. Similar to other EDCs, genistein produces a biphasic response in target cells. For example, depending on the concentration of genistein in the plasma of individuals consuming different amounts of soy dietary products (including soy supplements), cardioprotective (even if controversially reported) or cardiotoxic effects should be expected. The latter effects are related to much higher concentrations of genistein in the plasma (1–10 µM versus <1 µM) that produce potent inhibition of many membrane and cytosolic tyrosine kinases by competitively binding the ATP-binding sites of these kinases [138,139]. Soy PEs can also adversely affect thyroid function in susceptible individuals because in vitro studies have demonstrated that these compounds inhibit thyroid peroxidase (TPO), which is an enzyme involved in the synthesis of triiodothyronine (T_3_) and thyroxine (T_4_) [140,141]. In clinical settings, it has been established that patients with subclinical hypothyroidism receiving PEs in the diet are at higher risks of developing the overt form of the disease [142]. However, even with a higher utilized dose, a later study by the same team of researchers failed to confirm these findings [143]. In the most recently published study on rats, the consumption of relevant doses of soy isoflavones during the peripubertal period in males induced subclinical hypothyroidism, with alterations in the regulation of the hypothalamic–pituitary–thyroid axis, the modulation of thyroid hormone synthesis, and peripheral alterations in thyroid hormone target organs being observed [144].

Genistein may also adversely affect fetoplacental development. It has been proposed that the fetoplacental growth disruption pathomechanism of genistein involves its interference with placental growth factor (PlGF) signaling [145]. In vitro data have shown that both genistein and daidzein may bind to uterine ERs and induce either anti-estrogenic or weak estrogenic effects (higher and lower concentrations, respectively), thus influencing uterine responsiveness to oxytocin (OT) and prostaglandin F2-alpha (PGF2-α) and the corresponding contractility of the uterus [146]. The results of studies on human term trophoblast cells in vitro have shown that genistein and daidzein sufficiently reduce progesterone production in trophoblast cells via the disruption of estrogen receptor activity. Given that the blockade of progesterone is a possible mechanism involved in the initiation of labor, high doses of PEs at the feto-maternal unit could play a negative role in the maintenance of pregnancy. The compensatory mechanism observed in response to these PEs included higher estrogen production by trophoblast cells [147]. The clarification of whether a phytoestrogen-rich diet in pregnancy may pose an increased risk of preterm uterine contractions and subsequent preterm delivery requires further investigation.

Genistein exposure of infants may occur at physiologically relevant concentrations in the human diet that can be reached by using soy-based infant formulas. Infants consuming these products have serum genistein levels that are almost 20 times greater than those seen in vegetarian adults [148,149]. Importantly, the much weaker estrogenic activity of PEs can be compensated for by their high concentration in the body. For example, infants on soya formula can have plasma levels of isoflavones as high as 1000 ng/mL, which is 13,000–22,000 times higher than their own endogenous estrogen levels, as well as 50–100 times higher than estradiol levels in pregnant women and approximately 3000 times higher than estradiol levels at ovulation [132,150,151]. Consistently, plasma isoflavone levels in infants fed cow’s milk formula or human breast milk were much lower (9.4 and 4.7 ng/mL, respectively) than those in soy-based infant formula consumers [132,149]. To date, there have been no extensive studies on the potential endocrine-disrupting adverse effects of soya products in infants; however, the problem should not be ignored. Most of the recent animal studies have shown that comparable exposures have adverse physiological effects [152]. A previous study on mammals has shown that individuals from a population subjected to high consumption of isoflavones developed alterations in characteristics that may be of importance from an evolutionary perspective, such as epigenetic and morphometric characteristics or sexual maturation, which represents a life history characteristic [153]. It is likely that the most severe effects of hormonal disruption occur especially during a steroid-hormone-sensitive period termed “minipuberty” when estrogenic chemical exposure (including isoflavone exposure) may alter normal reproductive tissue patterning and function [154]. Minipuberty is the transient sex-specific activation of the hypothalamic–pituitary–gonadal (HPG) axis during the first 6 months after birth in boys and during the first 2 years in girls. During the course of this important genital organ development period, increases in luteinizing hormone (LH), follicle-stimulating hormone (FSH), E_2_, and testosterone are observed [153,154]. There are more data supporting the hypothesis that the disruption of development during this infant period in females may increase the risk of endometriosis in adulthood [153]. Moreover, developmental exposure to PEs may promote sensitivity to estrogen signaling diseases, including uterine fibroids and endometriosis. According to the results of population studies on soy phytoestrogen exposure, especially endometriosis, an estrogen-driven disease may have a developmental origin. In a study of 340 females diagnosed with endometriosis and 741 endometriosis-free, population-based controls, infant soy formula consumption was associated with over twice the risk of developing endometriosis relative to unexposed females [155,156]. The soy-formula-exposed group was even at a higher risk of developing endometriosis compared to gestational parental exposure to diethylstilbestrol (DES), which is the compound with endometriosis induction efficacy that has been demonstrated in several epidemiological and animal studies [157,158,159]. There is still a need to understand the molecular mechanisms and to investigate how PEs can influence epigenetic patterns during development.

Due to its prevalence and well-known estrogen-like effects, another family of dietary EDCs produced by fungi called mycoestrogens should be mentioned. The compounds known as mycotoxins are found in poorly stored cereals. For example, natural products with estrogenic activities found in *Fusarium crookwellnese* (syn. *Fusarium cerealis*) include zearalenone, alpha-trans-zearalenol, beta-trans-zearalenol, fusarin, fusarenone X, and nivalenol [160,161]. Zearalenone, which is a mycotoxin with a structure similar to that of naturally occurring estrogens, consists of a resorcinol moiety fused with a 14-member macrocyclic lactone and is the best-known representative of this group of EDCs [162]. Exposure to zearalenone and fusarin C has been linked to increased cancer rates. In in vitro studies, both fusarin C and zearalenone and its metabolites could stimulate the growth and proliferation of human breast tumor cells [163,164]. In addition, in vivo exposure of rats to environmental doses of zearalenone in the last two to three weeks of fetal development and in the first days after birth resulted in long-term changes in the development of the mammary gland, which was also associated with increased risks for the development of mammary tumors [47,165]. The ingestion of a sufficiently high dose of zearalenone in the diet may pose a risk to human health, not only because of its genotoxicity but also because of other adverse effects, including reprotoxicity and oxidative stress [166,167,168].

The results of studies on the involvement of zearalenone and other estrogenic mycotoxins, as well as Pes, in the etiopathogenesis of endometriosis, are ambiguous [164,167,168,169,170]. In conjunction with the ability of PEs to induce anti-proliferative, anti-inflammatory, and proapoptotic effects on cultured endometrial cells, beneficial effects have been reported in in vitro studies related to the inhibition of the spreading of endometriotic foci [46]. It has been proposed that this in vitro action of PEs involves the alteration of cell cycle proteins, the activation/inactivation of regulatory pathways, and the modification of radical oxidative species levels [47,171]. However, in the case of zearalenone, a dual role and opposite effects on endometrial cells may be observed, which is dependent on the estrogen concentrations in the environment. Therefore, zearalenone acts as an antagonist and an inducer of apoptosis in endometriotic tissue when estrogen is sufficient; however, it transitions to estrogenic activity in the absence of estrogen during the development of endometriosis [170]. The results derived from animal models of endometriosis have generally supported the beneficial effect of PEs in reducing lesion growth and development [169,171]. However, it is significant that the large amount of in vitro and in vivo animal findings did not correspond to consistent literature regarding the women affected with endometriosis. Therefore, whether the experimental findings can be translated to women is currently unknown [47,159,169].

When regarding the etiopathogenesis of endometriosis, it may be important that endocrine disruption through GPER is linked to rapid epigenetic effects because the heritable, regulatory elements of a genome (exclusive of its primary DNA sequence) play an essential role in maintaining the correct, undisturbed development of the organism and influence its homeostasis [172,173]. Recently, evidence has emerged that epigenetics appears to be a common denominator for hormonal and immunological aberrations in endometriosis [174,175]. Moreover, the regulation of expression of all known estrogen-responsive and progesterone (P4)-responsive receptor types by epigenetics may be a critical factor for endometriosis [176].

#### Endocrine Disruption and Altered Immune Function

Interactions of PEs with estrogen receptors that correspond to endocrine disruption may influence any aspect of hormone action. It is becoming increasingly clear that EDCs (including Pes) not only affect endocrine function but also adversely affect immune system function [177]. Importantly, in endometriosis, which is an estrogen-dependent and progesterone-resistant chronic inflammatory disease, the immune system fails to recognize and target endometrial tissue growing in ectopic locations (outside of the uterine cavity) in the body. This failure may indicate that endometriosis is an immune disease [178,179].

In general, PEs can suppress the immune response both in vivo and in vitro. This effect is due to their ability to inhibit nuclear factor kappa-light-chain-enhancer of activated B cell (NF-κB) intracellular signaling pathways [180,181]. NF-κB is a crucial transcription factor that participates in a number of physiological and pathological conditions, including the immune response, apoptosis, carcinogenesis, and inflammatory processes [182]. PEs (e.g., genistein) can suppress specific immune responses and lymphocyte proliferation [183]. Additionally, genistein can inhibit an allergic inflammatory response. In studies on mice, it has been shown that the administration of genistein in the diet produces reversible 46–67% decreases in the delayed-type hypersensitivity response, with reduced cell infiltrations in genetically treated animals compared with controls [184]. Genistein and daidzein, in particular, can suppress allergic inflammation by significantly reducing (by 25–30%) mast cell degranulation [185,186]. Consistently, the numbers of CD4^+^ and CD8^+^ T cells in normal lymph nodes were reduced in histopathological examinations. In contrast, it was demonstrated that genistein can increase cytokine production from T cells and enhance cytotoxic responses mediated by natural killers and cytotoxic T cells [187]. The treatment of activated dendritic cells (DCs) with genistein or daidzein led to increased NK-cell degranulation and cytotoxicity. This increased NK cell cytotoxicity was not influenced by other effects mediated by Pes, including the reduced expression of IL-18 receptor alpha (IL-18Rα) and the decreased production of interferon gamma (IFN-γ) in response to IL-12 and IL-18 [188].

Many studies have demonstrated that isoflavones and coumestrol can decrease the serum level of immunoglobulin G2a (IgG_2a_) antibodies. During experimental thyroiditis, low-dose coumestrol was able to decrease the titers of antigen-specific IgG_1_ and IgG_3_. Other isoflavones were effective in the suppression of IgE, thus possibly participating in the formation of the overall anti-allergic phenotype. Such a phenotype has been described in animal models, including airway and peanut sensitization models [185].

The vast majority of independent research has also demonstrated modulation concerning the inhibition of the innate immune system under the influence of PEs. Genistein, daidzein, and glycitein are able to inhibit the production of IFN-γ, tumor necrosis factor alpha (TNF-α), and interleukins IL-9 and IL-13 by CD_4_^+^ T cells in response to interaction with DCs. Direct cytokine secretion from activated DCs was also inhibited by these PEs [189]. It was also shown in an intranasal allergic response model that PEs may temporarily block the cell surface expression of major histocompatibility complex class I (MHCI) (but not MHCII) molecules during the maturation of DCs. Thus, a significant delay in the immune response caused by altered antigen-presentation and effector-cell priming functions of DCs should be expected [185,188]. The anti-inflammatory action of PEs in DC lines is still under investigation, in conjunction with the dual response (either pro-inflammatory or anti-inflammatory) that can be observed in NK cells.

Given that classically activated macrophages are products of a cell-mediated immune response, the proven anti-inflammatory phytoestrogen performance may be due to the fact that they make the full spectrum of macrophage activation more difficult [189,190]. Genistein and daidzein can decrease the synthesis of nitric oxide and the expression of inducible nitric oxide synthase (iNOS) with the accompanying increase in superoxide dismutase and catalase activities. Moreover, it has been demonstrated that genistein administration may alter macrophage polarization toward the noninflammatory M2 phenotype with a subsequent decrease in inflammatory cytokine concentrations [191]. M2 macrophages are necessary for the regulation of the resolution phase of inflammation and the repair of damaged tissues. In addition, genistein produces a strong expression of interleukin 10 (IL-10) in macrophages, which can limit the host immune response to pathogens, thereby preventing damage to the host and maintaining normal tissue homeostasis [192].

The complex action of PEs in relation to the innate immune system may explain the well-documented systemic anti-inflammatory effects of these xenoestrogens, including decreased allergic responses and decreased autoreactive immune responses [183,184,193]. The consumption of soy is growing at a significant rate, and its immune effect is extended. As the immune system influences basic physiological processes, including metabolic health, it seems likely that evolutionary alterations will be observed. It is important to monitor this situation and, if necessary, to prevent possible long-term detrimental consequences because quantitatively or qualitatively enormous amounts of PEs may cause pathological and epigenetically inherited alterations/dysfunction to the immune system.

## 2. Endometriosis

### 2.1. General Characteristics of the Disease

The name “endometriosis” refers to the condition in which endometrial tissue grows outside of the uterine cavity [194]. Depending on the location of the endometriotic foci, an endopelvic or extrapelvic form of endometriosis can be distinguished [195]. Abnormally implanted endometrial tissue is primarily found in the pelvis, including the ovaries, ovarian fossa, fallopian tubes, uterine wall (endometriosis genitalis interna or adenomyosis), broad ligaments, round ligaments, uterosacral ligaments, appendix, large bowel, ureters, bladder, or rectovaginal septum [194,196,197]. Extrapelvic localization of endometriosis is uncommon, and the disease is still underdiagnosed. Nevertheless, several cases of endometriosis of the upper abdomen, abdominal wall, abdominal scar tissue, diaphragm, pleura, pericardium, liver, pancreas, and lower and upper respiratory tract tissues (or even brain) have been reported [198,199,200].

Endometriosis affects 10–15% of women between the ages of 15 and 44 years and is associated with chronic pelvic pain, dysmenorrhea, dyspareunia, and infertility. Endometriotic foci contain tissue that is virtually the same in terms of biological properties as basal intrauterine endometrial tissue [201]. This tissue contains stromal cells, glands, and smooth muscles and is innervated and vascularized, with the presence of blood and lymphatic microvessels [201,202]. The cells within endometriotic lesions express all of the receptors for estrogens (Erα, Erβ, and GPER) and progesterone (PR-A and PR-B). Therefore, they react to hormonal changes during the menstrual cycle and are subjected to cyclical changes analogous to the endometrium, ranging from re-epithelization and proliferation to breakdown and desquamation. In the uterine cycle, this corresponds to the phases of proliferation, secretion, and menstruation [203,204]. The lack of blood outflow from the extrauterine “trapped” endometrial cells may predispose patients to internal bleeding that remains on site. Such bleeding may be the starting point of the local inflammatory response, accompanied by pain and the development of more serious fibrosis-based complications [205]. Due to pain, the quality of life of women suffering from endometriosis may be significantly compromised. Additionally, fibrosis and scarring with the formation of adhesions will be elicited as a result of repair processes within inflamed endometriotic tissue and its vicinity [194,199,205]. The question that needs to be resolved is whether the inflammatory process favors the development of endometriosis foci or whether endometriosis foci induce the inflammatory process [206,207]. In addition to pain-related dysmenorrhea and dyspareunia, the disease makes it difficult to get pregnant and to have a successful pregnancy outcome [208,209]. The disturbance of reproductive potential in endometriosis is partly due to the intensification of the cell senescence process, which is accompanied by chronic inflammation, referred to as inflammaging [210,211]. Moreover, a higher incidence of cancer and autoimmune diseases has been linked to endometriosis [212].

Despite several decades of intensive investigation into the underlying etiology and pathogenesis of endometriosis, the current understanding of the disease remains unclear. Several theories for the pathogenesis of endometriosis have been elaborated or updated in recent years, including implantation (retrograde menstruation) and metaplasia of Müllerian-type epithelium (coelomic metaplasia) theories, as well as the induction theory (a combination of the previous two theories), which emphasizes the impact of unidentified substances released from shed endometrium that induce the formation of endometriotic tissue from undifferentiated mesenchyme [213,214]. The implantation theory has been supplemented with new data indicating that the endometrium contains a particular population of cells with clonogenic activity that resembles the properties of mesenchymal stem cells, in which the dysfunction of these cells may lead to the formation of initial endometrial lesions [215].

It has recently been shown that the increased activity of myeloid-derived suppressor cells (MDSCs) promotes ectopic growth in endometriosis [216]. MDSCs are a heterogeneous population of immature myeloid cells (dendritic cells, granulocytes, and monocyte/macrophage precursors), which play an important role in the development of immunological diseases, such as chronic inflammation and cancer, due to their ability to selectively suppress both innate and adaptive immune responses [216,217].

It has also been proposed that stem cells derived from bone marrow may be a primary source of endometriotic cells [218,219]. The most recent hypothesis suggests that endometriosis risk is driven by relatively low levels of prenatal and postnatal testosterone. Testosterone affects the developing hypothalamic–pituitary–ovarian (HPO) axis; moreover, at low levels, it can result in an altered trajectory of reproductive and physiological phenotypes that, in extreme cases, can mediate the symptoms of endometriosis [220]. In summary, endometriosis is a multifactorial disease with the involvement of genetic, immunological, hormonal, anatomical, and environmental factors in different proportions [206,207] (Figure 3).

### 2.2. Disruption in Estrogen and P4 Signaling

Hormone release dynamics and the interplay between the main female sex steroid hormones, including estradiol (E_2_) and progesterone (P4), govern the periodic growth and regression of the endometrium. Thus, such a balance between E_2_- and P4-responsive signaling pathways creates an extraordinary environment for controlled tissue remodeling during the menstrual cycle. In normal endometrium, where estrogen and P4 signaling coordination is tightly regulated, this remodeling plays a key role in decidualization to allow for implantation during the window of receptivity, as well as, in the absence of fertilization, for the disintegration of the endometrium, thus leading to menstruation [221].

According to the implantation theory of endometriosis, which assumes the spreading out of endometrial stromal cells (EnSCs) with the menstrual blood to establish ectopic growth (endometriotic foci), there is a significant disruption in estrogen and P4 signaling, which commonly results in P4 resistance and E_2_ dominance [222]. Thus, a hormonal imbalance caused by the actual or relative excess of E_2_ throughout the menstrual cycle and the expression of their cognate nuclear receptors, the progesterone receptors (PR-A and PR-B), and estrogen receptors (ERα and ERβ) deserves attention [223]. Moreover, the mutual affinity of nuclear receptors for the main female sex steroid hormones is necessary, given that the interaction of two domains of the P4 receptor with ER is required for P4 activation of the proto-oncogene tyrosine-protein kinase Src/extracellular signal-regulated kinase (c-Src/ERK) pathway in mammalian cells [224]. Additionally, sex steroid membrane receptors that are responsible for rapid nongenomic signaling/responses have garnered attention, also in the context of endometriosis. It has been demonstrated that P4 affects cell proliferation and survival via nongenomic effects. In this process, membrane progesterone receptors (mPRα, mPRβ, mPRγ, mPRδ, and mPRε) were identified as being putative G protein-coupled receptors (GPCRs) for progesterone [225]. Similarly, the G protein-coupled estrogen receptor (GPER) is a seven-transmembrane-domain receptor that mediates nongenomic estrogen-related signaling. After ligand activation, GPER triggers multiple downstream pathways that exert diverse biological effects on the regulation of cell growth, migration, and programmed cell death in a variety of tissues, including the human endometrium [109,204].

It is worth noting that chronic stress and inflammation also lead to a further imbalance between P4 and estrogen, thus exacerbating the course of preexisting endometriosis [226].

#### 2.2.1. Estrogen Dominance

The symptoms of estrogen excess and estrogen dependence in endometriosis are striking. This observation is limited to endometrial tissue and ectopic endometrial foci because the intratissue estrogen concentrations do not reflect the corresponding serum levels [223,227].

Absolute or relative hyperestrogenism, which is well documented in endometriosis, can also confirm the fact that estrogen-dependent endometriosis is rarely diagnosed after menopause when the symptoms and endometriotic lesions are typically relieved [228]. Similarly, during pregnancy, when estrogen action is oversuppressed by the influence of P4 or while taking hormonal contraceptives (e.g., via the use of ethinylestradiol-containing pills) that cause pharmacological suppression of endogenous estrogen synthesis, the severity of the disease usually decreases [229,230].

##### Aromatase Activity

Aromatase (EC 1.14.14.1), which is also known as estrogen synthetase or estrogen synthase, is a unique rate-limiting enzyme that transforms androgen precursors into estrogens via aromatization. This member of the cytochrome P450 family (CYP) and the product of the CYP19A1 gene is responsible for the conversion of androstenedione, testosterone, and 16-hydroxytestosterone into estrone (E_1_), estradiol (E_2_), and estriol (E_3_), respectively [231]. The most potent endogenous estrogen E2 exhibits extremely strong mitogenic properties in endometriotic tissue. Hence, any alterations in aromatase activity will produce a shift in the balance between estrogenic and androgenic effects within responsive tissues. Not coincidentally, the growth of ectopic endometrial tissue requires high aromatase activity induction, which is normally not detectable in eutopic (located in the proper place as the inner lining of the uterus) endometrium [232]. In contrast to normal endometrium, where estrogens are not locally produced, endometrial stromal cells (EnSCs) isolated from women with pelvic endometriosis exhibit significantly high P450 aromatase mRNA expression levels [233].

Analogous to breast cancer, abnormally expressed aromatase in EnSCs within endometriotic foci may be stimulated by prostaglandin E_2_ (PGE_2_) via the promoter II region of the aromatase gene. When considering the fact that PGE_2_ is one of the best-known mediators of inflammation and pain, the local production of estrogens will be accompanied by the typical pain of the disease. Moreover, a positive feedback loop (aromatase-PGE_2_-aromatase) is established because estrogen itself upregulates cyclooxygenase 2 (COX-2) and subsequently stimulates PGE_2_ formation [231,232,233].

It has been documented that the hyperestrogenic nature of the microenvironment within endometriotic lesions is the derivative of an epigenetic regulatory mechanism action involving the aromatase gene (CYP19A1), which is located on chromosome 15q21. Thus, endocrine disruption by dietary PEs may be important as an epigenetic modulator of estrogen signaling at the level of endometrial foci. Additionally, multiple exons of CYP19A1 may be alternatively used in endometriotic cells corresponding to EnSCs that exploit identical aromatase promoters (promoters II, I.3, and I.6) as aromatase-negative eutopic endometrial cells [175,234,235]. Given that endometriotic stromal cells are equipped with the same set of promoters as normal eutopic EnSCs, the differences in aromatase gene expression may be caused by an epigenetic regulatory mechanism that inhibits aromatase gene expression in healthy endometrium, whereas this effect is not present in endometriosis. The confirmation of the abovementioned effect may be the fact that CpG islands (the regions of the genome that are rich in promoters) are hypomethylated in endometriotic cells and hypermethylated in endometrial cells [236]. DNA methylation is strictly linked to histone modifications and the recruitment of histone deacetylases (HDACs), followed by chromatin condensation. It is generally accepted that hypomethylated genes possess an increased potential for expression compared to hypermethylated genes [237]. Thus, the differential expression of the aromatase gene between normal intrauterine endometrium and endometriotic foci may be due to the absence or presence, respectively, of the transcription factor known as steroidogenic factor 1 (SF-1). It has been found that methylation of CpG islands in the SF-1 gene, which spans from exon II to intron III, positively regulates its expression in EnSCs in endometriosis, whereas hypomethylation of SF-1 gene CpG islands in eutopic endometrium drastically decreases SF-1 levels [238,239].

Deficient 17β-hydroxysteroid dehydrogenase type 2 (17β-HSD2) expression is another abnormality that has been reported in endometriosis, and it predisposes afflicted individuals to hyperestrogenism. Normally, the accumulation of increasing quantities of E_2_ in target tissues is counteracted by the conversion of adequate levels of 17β-estradiol to much less potent estrone (E_1_) [240]. This pathway of E_2_ inactivation is disrupted in ectopic EnSCs via hypermethylation of the 17β-HSD2 gene, thus resulting in insufficient 17β-HSD2 activity within endometrial lesions [241]. Although of unknown importance in endometriosis, it should be mentioned that the same epigenetic mechanism (e.g., DNA methylation) is likely to influence the activity of 17β-hydroxysteroid dehydrogenases type 1 and 4 (17β-HSD1 and 17β-HSD4, respectively), which are enzymes present in the human endometrium and EnSCs [242,243].

All of these interrelationships between epigenetic modulators of aromatase activity and hyperestrogenism are summarized in Figure 4.

#### 2.2.2. The Importance of Epigenetic Factors

The epigenome is defined as the complete description of all of the chemical modifications to DNA and histone proteins that regulate the expression (activity) of genes within the genome without interfering with the DNA nucleotide sequences, and it encompasses both small and long noncoding RNAs (miRNAs and lncRNAs, respectively) [244,245]. Epigenetic changes occur regularly and naturally in response to aging, the environment/lifestyle, and disease states. Furthermore, this phenomenon aims to maintain genomic integrity [246,247].

The properties of cellular targets for epigenetic factors in endometriosis are very particular because EnSCs with clonogenic potential constitute the most abundant population of cells within the endometrium and endometriotic tissue that resemble the properties of mesenchymal stem cells (MSCs) [248]. The unique nature of stem cells involves the ability to divide and renew themselves for long periods of time, as well as unspecialization and the capability of differentiating into specialized cell types [249]. Therefore, stem cell plasticity causes the precise control of both metabolism and gene expression to be rapidly adjusted to varying conditions (e.g., hormonal status and the phase of the menstrual cycle), including environmental factors related to dietary intake of PEs and other compounds with endocrine-disrupting potential [169,175,250,251].

The failure of epigenetic homeostasis in the endometrial tissue may demonstrate local intrauterine abnormalities or a generalized systemic disorder during repeated menstrual cycles or pregnancies [219,252]. Research results from recent years have determined that the regulation of ERs and P4 receptor expression by epigenetics may be a critical factor for endometriosis [176,253,254]. Specifically, disrupted estrogen and P4 signaling that correspond to increased estrogen activity and P4 resistance, respectively, are the main substrates of the disease, wherein environmental factors contribute to the inflammatory response and debilitating symptoms, including pain and infertility.

##### Epigenetic Modulation of ERs in Endometriosis

It has been demonstrated that ERs in EnSCs are subjected to the same epigenetic regulation as in other estrogen-reactive tissues [239,255,256]. In human endometriotic stromal cells corresponding to EnSCs, markedly higher levels of ERβ and lower levels of ERα have been reported compared to EnSCs obtained from eutopic endometrium [257,258]. Such overexpression of ERβ in endometriosis has been linked to significantly pathologically reduced methylation of a CpG island in the promoter region of the ERβ gene (*ESR2*). Conversely, bisulfite sequencing of this region has identified significantly higher methylation in primary endometrial cells versus endometriotic cells [259]. Consequently, the experimental use of a demethylating agent can significantly increase ERβ mRNA levels in endometrial cells. Moreover, the overexpression of ERβ in endometriosis correspondingly suppresses ERα expression and response to E2 in EnSCs by binding to nonclassical DNA motifs in alternatively used ERα promoters [203]. Thus, the normal response pertaining to ERα expression in endometriotic lesions is suppressed by both abnormally high quantities of E_2_ resulting from local aromatase overactivity and the epigenetic upregulation of ERβ in stromal cells [260]. When considering that the P4 receptor (PR) gene is induced in reproductive tissues by estrogen acting via ERα, the decreased expression of ERα observed in endometriosis may contribute to P4 resistance, which is a typical feature in women suffering from this disorder [203,261].

The proliferation of endometriotic lesions can also be linked to severely increased ERβ mRNA levels in EnSC- and/or MSC-derived endometriotic cells following DNA demethylation because ERβ signaling stimulates cell cycle progression [262].

Extraordinarily higher ERβ and significantly lower ERα and PR expression in endometriotic stromal cells compared with endometrial stromal cells may be caused by another epigenetic mechanism related to small (19–25 nucleotides long), single-stranded noncoding RNAs (miRNAs) that regulate gene expression. This dominant pool of RNA does not code for proteins but is processed to produce functional RNAs, and miRNAs are crucial regulators of gene expression in E_2_-treated human endothelial cells [263,264].

Based on animal models and human studies, ER expression during the different phases of the menstrual (endometrial) cycle is modulated by miRNAs [263,265]. These data relate especially to the numerous miRNAs that directly target ERα, whereas less information is available for miRNAs modulating ERβ and GPER [266,267,268,269].

Nevertheless, results indicating that GPER-mediated downregulation of miR-148a expression through the GPER/miR-148a/HLA-G signaling pathway may mediate the development of ovarian endometriosis have recently been published [270]. In addition, the epigenetic regulation of ER expression by miRNAs coexists with opposing mechanisms that act in parallel, such as the ER-mediated regulation of miRNA expression. For example, E_2_-treated human umbilical vein endothelial cells (HUVECs) have differentially regulated specific miRNAs via pathways related to both classical ERs (ERα and ERβ) and membrane-bound ERs (GPER) [264]. Among the most modified miRNAs, miR-30b-5p, miR-487a-5p, miR-4710, and miR-501-3p were overexpressed after E_2_ treatment, whereas miR-378 h and miR-1244 were downregulated [264].

In addition to miRNAs, researchers studying the epigenetic regulation of estrogen signaling have recently focused on the role of some transcripts longer than 200 nucleotides that lack protein-coding potential and transcribed by RNA polymerase II (RNA Pol II), which are known as long noncoding RNAs (lncRNAs) [271]. Together with the research progress on lncRNAs, there is increasing evidence that by regulating the epigenetic status of protein-coding genes, lncRNAs are involved in the pathogenesis of endometriosis [272]. For example, the upregulation of lncRNA HOTAIR is caused by E_2_ binding to ERα and ERβ. Moreover, coregulators, including histone methyltransferases (MLL1 and MLL3) and histone acetylases in the p300–CBP family, are recruited together with ERs to bind estrogen response elements in the HOTAIR promoter in response to E_2_; additionally, they are necessary for the upregulation of HOTAIR [273].

As was previously mentioned (see Section 1.1.1), estrogen signaling involves the recruitment of many coregulator proteins (coactivators and corepressors) that interact with many members of nuclear-receptor-related multifunctional protein complexes, thus resulting in both transcriptional and epigenetic changes. The latter changes include (but are likely not limited to) chromatin density changes, histone modifications by acetylation/deacetylation, and DNA methylation/demethylation, as well as noncoding RNAs. Therefore, the expression of ERs in health and disease may depend on the recruitment of comodulators that are crucial for the activities of the respective acetyltransferases (e.g., p300-CBP and its paralog, p300; GNAT or GCN5-related N-acetyltransferase, nuclear receptor coactivator-NCOA-related histone acetyltransferase) and methyltransferases (e.g., histone lysine N-methyl-transferases and histone arginine N-methyltransferases) [274,275,276].

Interestingly, being classified as an lncRNA, steroid receptor RNA activator (SRA), which acts as the nuclear receptor coactivator, can influence the activities of both ERα and ERβ [277]. High expression levels of SRA lncRNA and ERβ (but relatively low expression levels of SRA and Erα) have been demonstrated in ovarian endometriotic tissues compared to normal endometrium. In conjunction with the abovementioned findings, SRA1-small interfering RNA treatment significantly increased ERα levels but reduced ERβ levels in EnSCs. Such treatment with interfering RNA reduced proliferation within ovarian endometriotic foci and promoted the early onset of apoptosis in endometriotic cells [278].

ER activity may be regulated by sirtuins (SIRTs), which possess histone deacetylase (HDAC) activities and act as comodulators of both estrogen-regulated gene silencers and inhibitors of ligand-dependent activation of ERα [279]. The overexpression of SIRT1 may contribute to both the pathomechanism of endometriosis and P4 resistance [280]. Interestingly, eutopic end ectopic endometrial tissues obtained from the same patient differ in the content of SIRT1. Significantly decreased levels of SIRT1 mRNA were demonstrated in eutopic EnSCs compared to EnSCs from endometriotic lesions [281].

When considering that complex and nonuniform mechanisms of estrogen/ER signaling within endometrial cells are subjected to significant modulation by epigenetic factors, endocrine disruptors may induce pathological regulatory mechanisms that are responsible for ectopic EnSC persistence and the development of endometriotic foci [282,283,284,285].

### 2.3. Estrogen-Dependent Immune System Interactions in Endometriosis

First, the immune system is responsible for eliminating cells that are located in ectopic sites (endometriotic foci). The failure of this elimination in endometriosis may be due to both resistance of ectopic cells to be eliminated by immune cells and a deficit in the immune response [209,213]. Numerous studies have demonstrated that endometriosis is associated with aberrant growth and loss of sensitivity to apoptosis of endometrial tissue cells [179]. This effect may be confirmed by an increase in the expression of anti-apoptotic proteins, such as Bcl-2, c-IAP1, and c-IAP2, in ectopic endometrial cells compared to eutopic endometrial cells [286]. Thus, apoptosis-inducing processes that are mainly related to interactions with immune cells (e.g., cytotoxic T lymphocytes (CTLs), also known as killer T cells) may be suppressed, thus promoting the survival and development of endometriotic lesions [287,288]. Estrogen excess observed in endometriosis can activate both epithelial and stromal cells that constitute the population of endometriotic cells, thus causing the anti-apoptotic status of the respective ectopic tissue [179,289]. This scenario is facilitated by the impact of estrogen excess on CD4 T-helper development and function, especially with regard to the profile of the produced cytokines [290]. The immunosuppressive functions of Tregs are widely acknowledged and have been extensively studied [291,292]. Altered CD4 T lymphocytes may lead to disturbances in the coordination of the immune response by inappropriately stimulating other immune cells, such as macrophages, B lymphocytes (B cells), and CD8 T lymphocytes (CD8 cells), to fight ectopic endometrial foci development [178,291,293].

It should be noted that at the current stage of research, it is not possible to distinguish to what extent observed alterations are intrinsic to the endometriotic cells or are induced by their ectopic location [288,294,295]. Moreover, it has been demonstrated in previous studies on cancer cells that estrogen acting through different ER isoforms can induce opposing mechanisms (i.e., antiapoptotic types that promote tumor growth and proapoptotic types that promote programmed cell death). Accordingly, it has been shown that the E_2_/ERα complex activates multiple pathways involved in both cell cycle progression and apoptotic cascade prevention, whereas the E_2_/ERβ complex in many cases directs the cells to apoptosis [296].

Excess estrogen has a strong effect on the immune response because the immune system is a natural target for these classes of sex steroid hormones, and immune cells express all types of currently known receptors [297]. Although the cause of sex differences in the immune system has not been definitively identified, possible causes should be investigated, including different sex hormone profiles (estrogens, androgens, and differential sex-hormone-receptor-mediated pathways), X-chromosomes, microbiome, and epigenetic factors. Females tend to have a more responsive and powerful immune system than members of the opposite sex. The consequence of the abovementioned scenario is a more aggressive response to self-antigens and a more frequent prevalence of autoimmune diseases among women [298,299]. For example, extremely higher estrogen concentrations in females compared to males drive increased T-cell IFNγ production and, in this manner, predispose females to IFNγ–mediated autoimmune conditions [300]. To date, clinicians do not consider endometriosis an autoimmune disease; however, it resembles an autoimmune condition in many aspects [179,301].

It has been well established that E_2_ signaling participates in the precise control of proinflammatory-signal/pathway-related phenomena of the immune system [302,303,304,305]. Estrogen regulates key genes that are responsible for the innate and adaptive immune systems, and the list of immune cells that are subject to this regulation is almost complete, including granulocytes (neutrophils), monocytes (macrophages and monocyte-derived dendritic cells), and lymphocytes (T cells and B cells) [297]. For example, within the innate immune response, estrogen signaling modulates neutrophil numbers, migration, infiltration, and activation via genes coding cytokine-induced neutrophil chemoattractant proteins 1-3 (CINC-1, CINC-2, and CINC-3), TNFα, IL-1ß, and IL-6 [306,307,308]. In contrast, in macrophages, estrogen signaling may modify chemotaxis, phagocytic activity and induction of cytokines, iNOS, and nitric oxide by affecting genes IL-6, TNFα, iNOS, and NO, respectively [309,310,311,312]. In terms of the adaptive response, estrogen signaling modulates all subtypes of T cells, including CD4^+^ (Th1, Th2, Th17, and Tregs) and cytotoxic CD8^+^ cells (CTLs) [297,313,314]. For example, this modulation pertains to genes encoding interferon gamma (IFNγ) in Th1 cells; IL-4 in Th2 cells; and FoxP3, PD-1, and CTLA-4 in Tregs [314,315,316,317,318,319]. Thus, there is no doubt that estrogen plays a major role in shaping T-cell responses. This action is observed independently of the direct disruptive effect on gene transcriptional programs of T cells and involves T-cell maturation, activation, and differentiation [319,320]. Moreover, B-cell (B lymphocyte) differentiation, activity, function, and survival are also highly dependent on estrogen, which can modify the expression of genes such as CD22, SHP-1, Bcl-2, and VCAM-1 [321,322]. In certain states, estrogen acting through either ERα or ERβ may contribute significantly to autoimmune disorders because a study on autoimmune mice subjected to estrogen demonstrated increased plasma cell and autoantibody-producing cell numbers [323]. However, signaling via ERα is crucial in altered cell maturation coexisting with autoimmunity [324].

Estrogens can indirectly inhibit NF-κB DNA binding, as they have been shown to inhibit IKK activation, increase IkappaB protein expression, and decrease its phosphorylation [325,326,327,328]. ERα and GPER1 signaling is commonly associated with anti-inflammatory phenotypes, whereas data on ERβ signaling are not consistent, thus indicating both anti-inflammatory roles similar to ERα and GPER1 and proinflammatory effects in the case of an increased ratio of ERβ [297,329]. It may be important in the context of endometriosis that 17β-estradiol signaling via overexpressed ERα may inhibit inflammatory activation mediated by NF-κB and JNK via PI3K/AKT [330]. However, it is likely that reported differences in the effects of estrogen on the immune system are related to the timing at which such effects are observed following estrogen exposure, as well as variations in the respective type of ER expression in various cells and during different physiological or pathological conditions [288,297,328].

During the menstrual cycle of healthy women, increased concentrations of cytotoxic (CD8+) T lymphocytes (CTLs) and HLA-DR- activated T cells were observed in peripheral blood during the luteal phase compared to the follicular phase. These fluctuations in the concentrations of cytotoxic and activated peripheral blood lymphocytes are not present during the menstrual cycle of women with endometriosis [331]. Moreover, there has only been a marked increase in Treg concentration in the peripheral blood of women with endometriosis, which was positively correlated with the serum levels of cortisol [331]. In addition, a significant reduction in the cytotoxic/proapoptotic potential of CTLs was demonstrated in endometriosis, wherein the number of perforin+ CTLs among CD8+ T cells in the menstrual effluent was decreased compared to healthy controls. Perforin is a glycoprotein mediator of cytolysis that is responsible for pore formation in cell membranes of target cells, thereby causing the initiation of programmed cell death [332,333]. Perforin mRNA levels correlate with the methylation status and accessibility of the promoter at the 5′ flanking region of its gene. Thus, the defective apoptotic process may be caused by DNA hypermethylation and changed chromatin structure that negatively affects perforin gene expression in T cells [179,334].

#### Estrogen and Mast Cells (MCs) in Endometriotic Lesions

MCs express estrogen (ERα, Erβ, and GPER) and P4 receptors (PR-A and PR-B) and further respond to these hormones, which causes changes in the MC cell number, distribution, and functional state in various tissues [335,336]. It should be noted that E_2_ is implicated in the immune response as an enhancer, including MC activation and the subsequent release of mediators stored in the secretory granules (degranulation) [337]. Among the ERs, GPER is responsible for the various running-fast nongenomic effects of estrogens, including the degranulation of MCs [338]. The activation and degranulation of MCs significantly modulate many aspects of physiological and pathological conditions in various settings. MC secretory granules are lysosome-like organelles that contain a large panel of preformed bioactive constituents, including lysosomal hydrolases (e.g., carboxypeptidase A, chymase, and tryptase), amines (histamine), cytokines (interleukin (IL)-1, IL-2, IL-3, IL-4, IL-5, IL-6, granulocyte–macrophage-colony-stimulating factor, interferon-γ (IFN-γ]), and tumor necrosis factor-α (TNF-α)), and proteoglycans (e.g., heparin) [339,340]. These mediators are responsible for many of the acute signs and symptoms of MC-mediated allergic inflammatory reactions, including edema, bronchoconstriction, and increased vascular permeability [340,341]. In addition, MCs are involved in angiogenesis, fibrosis, and pain, and a significant increase in MC numbers within endometriotic lesions has been demonstrated compared to matched eutopic endometrium from the same patients [342,343]. Furthermore, endometriotic tissue specimens demonstrate a significantly higher expression of stem cell factor (SCF), which is a potent growth factor critical for MC expansion, differentiation, and survival for MCs localized in connective tissue [344]. Following pretreatment with estrogen in mice, the endometriotic foci demonstrated a higher density of Alcian-blue-stained MCs. In patients with endometroid endometrial cancer, MC density was positively correlated with angiogenesis, as assessed by local microvascular density [345].

The abovementioned results indicate that the conditions that are characteristic of the disease (particularly, the abnormal hyperestrogenic/P4 resistant endocrine microenvironment within endometriotic lesions) promote the recruitment and differentiation of MCs. As a result, MCs may release a diverse spectrum of mediators that contribute to inflammation, chronic pelvic pain, and local angiogenesis, thus resulting in disease progression [343,344,346]. Due to the fact that MCs are very prevalent in endometriotic tissue, it has been proposed that this population of MCs represents a therapeutic target in endometriosis to assure better control of disease inhibition and symptom relief [347].

The multifunctional nature of MCs includes their involvement in the regulation of innate and adaptive immune responses. Increasing evidence has suggested that MCs play a regulatory role in inflammatory diseases (such as endometriosis) by regulating T-cell activities. In addition to serving as effector cells, MCs are able to induce T-cell activation, recruitment, proliferation, and cytokine secretion in an antigen (autoantigen)-dependent manner and to impact regulatory T cells [348,349].

Estrogen-dependent immune responses related to MCs in endometriotic foci are shown in Figure 5.

## 3. Dietary PEs and Endometriosis

### 3.1. PE Intake and the Risk of Endometriosis—Interactions at the Level of Gut Microbiota

The gut microbiota or gut microbiome consists of microorganisms, including bacteria, archaea, fungi, and viruses, living in a state of dynamic equilibrium in the digestive tract. The aggregate of all of the genomes of the gut microbiota is known as the gastrointestinal metagenome, and it is very large [351]. For example, the microbiota “organ” is the central bioreactor of the gastrointestinal tract, and it is populated by a total of 10^14^ bacteria and characterized by a genomic content (microbiome), which represents more than 100 times the human genome. Bacteria account for up to 60% of the dry weight of feces [352,353]. Due to this scenario, the symbiosis and dysbiosis of this dynamic ecosystem play an important role in health and disease, respectively [353]. Colonization of bacteria that make up the microbiome has a broad impact on resistance to pathogens, whereby it maintains the intestinal epithelium, metabolizes dietary and pharmaceutical compounds, controls immune function, and even (to some extent) controls behavior through the gut–brain axis [354,355,356]. Moreover, when considering the plasticity of the gut microbiota, diet has emerged as a main contributor to the microbiota composition and functional capacity. The number of studies showing that food/nutrient–microbiota interactions are important modulators of host physiology and pathophysiology is constantly increasing [357].

With respect to estrogens, as early as 2011, Plottel and Blaser proposed the estrobolome as the aggregate of enteric bacterial gene products that were capable of metabolizing estrogens [358]. The gut microbiota regulates estrogens through the secretion of β-glucuronidases [E.C. 3.2.1.31] by some bacterial species, an enzyme that deconjugates estrogens into their active forms, capable of entering enterohepatic circulation. It has been reported that, under physiological conditions, fecal β-glucuronidase is negatively proportional to the total estrogen levels in circulation. Thus, an excess of estrogens induces gut microbiome diversity, thereby decreasing β-glucuronidase availability and increasing estrogen excretion. Conversely, low estrogen levels decrease gut microbiome diversity, which increases β-glucuronidase activity and re-circulation of estrogens [359,360]. A diet that is rich in fat or protein has been associated with high fecal levels of β-glucuronidase, while a fiber-based diet decreases the activity of this enzyme [361]. In the gastrointestinal tract, the most important genes encoding β-glucuronidase enzyme activity are the β-glucuronidase (GUS) genes. More than 110 GUS genes have been identified and grouped into six classes expressed in four bacterial phyla, denominated as Bacteroidetes, Firmicutes, Verrucomicrobia, and Proteobacteria [362].

Accordingly, a phytoestrogen-rich diet affects the composition of the gut microbial community (gut homeostasis) as an environmental epigenetic factor and provides metabolites that influence host physiology, including endocrine balance and the potential risks of endocrine disruption [137]. The microbiota undoubtedly functions as a full-fledged endocrine organ influencing the reproductive endocrine system throughout a woman’s lifetime by interacting with estrogen, androgens, insulin, and other hormones [363]. For example, the mammalian PEs enterolactone and enterodiol are formed in the colon by the action of bacteria on plant lignans by matairesinol and secoisolariciresinol, which exist in various whole-grain cereals (barley, rye, and wheat), seeds, nuts, legumes, and vegetables. Both enterolactone and enterodiol have been shown to possess weakly estrogenic and antiestrogenic activities, and it has been suggested that the increased production of these antiestrogenic mammalian lignans in the gut may serve to protect against breast cancer in women and prostate cancer in men [364]. Furthermore, the protective effects of these mammalian lignans may be due to their ability to compete with E_2_ for ERs, as well as to induce sex-hormone-binding globulin (SHBG), to inhibit aromatase and act as antioxidants [365].

Due to the fact that the gut flora affects immune health by controlling inflammatory responses and plays an important role in estrogen metabolism and in the regulation of estrogen cycling, the onset and progression of endometriosis may be a consequence of gut dysbiosis. Such dysbiosis may modulate the estrobolome with a subsequent increase in the levels of circulating estrogen, which may markedly stimulate the growth and cyclic bleeding within endometriotic foci [366]. The results of recent preclinical and clinical studies suggest that altered interactions between gut permeability and intestinal (as well as extraintestinal) bacteria collectively contribute to systemic inflammation and metabolism. According to the “leaky gut” concept, disturbances in the composition of the intestinal microflora may change gut permeability and predispose to alterations in different types of immune cells and inflammatory factors (e.g., increased levels of inflammatory cytokines in the peritoneal fluid and serum) in endometriosis [367,368]. When referring to the pathomechanism of endometriosis, once the balance between estrogen levels in circulation and the gut microbiome is disrupted, increased estrogen exposure due to a phytoestrogen-rich diet can stimulate the development and progression of endometriotic lesions [369]. A strong interrelationship between immunological processes and endometriosis was confirmed by the observation that the risk of inflammatory bowel disease (IBD) in women with endometriosis increases by 50% [370]. In a murine model of endometriosis, dysbiosis of the gut microbiota was manifested by an elevated *Firmicutes/Bacteroidetes* ratio and an increased ratio of *Bifidobacterium* [371]. Another previous study in mice showed the effectiveness of broad-spectrum antibiotic therapy in limiting the inflammatory response and in inhibiting the growth of endometriotic tissue. Oral gavage of feces from mice with endometriosis restored endometriotic lesion growth and inflammation, thus indicating that gut bacteria may promote endometriosis progression in mice [372]. Additionally, a higher prevalence of intestinal inflammation coexisting with dysbiosis of gut microflora (lower lactobacilli concentrations and higher Gram-negative bacterial load) was also documented in experimental endometriosis in rhesus macaques (*Macaca mulatta*) [373]. In a previous human study, it was found that women with stage 3/4 endometriosis excrete more *Escherichia/Shigella* in their stool compared to the control group, whereas the vaginal, cervical, and gut microbiota compositions were similar [369].

In summary, the unmistakable connections between changes in the intestinal microbiome and gut homeostasis, intestinal permeability, and inflammation deserve to be pursued in future research in the context of estrogen excess in endometriosis [363,374].

### 3.2. PE Oral Intake and the Course of Endometriosis—The Results in Animal Models

The results of research obtained on animal models of endometriosis (with the exception of higher primates) should be treated with great reserve because the estrous cycle is not equal to the menstrual cycle; in addition, a typical, plant-based diet in rodents in the natural environment has a much higher phytoestrogen content than a typical nonvegan, nonvegetarian human diet [375,376]. Translational animal models for endometriosis developed in mice (female BALB/C) and rats (female Sprague Dawley or Wistar albino) conducted endometriotic cell transplantation into sites on the peritoneum or intestinal mesentery via intraperitoneal injections or homologous uterine horn transplantations [375]. Based on the review of the obtained results for different PEs, it can be concluded that these compounds that are orally administered can cause the regression of endometriotic implants.

**Resveratrol** is one of the most studied compounds. This nonflavonoid polyphenol that naturally occurs as a phytoalexin inhibited the development of experimental endometriosis in mice and reduced endometrial stromal cell invasiveness in vitro. Mice treated orally with resveratrol (6 mg/mouse; *n* = 20) for 18–20 days exhibited both statistically significant decreases in the number of endometrial implants and in the total volume of lesions (by 60% and 80% per mouse, respectively). It is worth noting that human endometrial stromal cells were used to induce endometriosis [377]. In another study on BALB/C mice with surgically induced endometriosis, resveratrol (40 mg/kg/day; *n* = 10) that was orally administered for 4 weeks inhibited angiogenesis in peritoneal and mesenteric endometriotic lesions, as indicated by a significantly reduced microvessel density when compared with controls. Decreased proliferating activity of CD31(+)-positive cells in the newly developing microvasculature of the lesions was also confirmed. This scenario coexisted with lower numbers of proliferating cell nuclear antigen- and Ki67-positive stromal and glandular cells. The authors noted limitations in translating the results into human conditions, which was caused by the mouse model that was used in the study [378]. Similar results have also been demonstrated in a study on the effects of resveratrol and another polyphenol known as epigallocatechin-3-gallate (EGCG) on the development of endometriosis in a BALB/C mouse model. Both treatments significantly reduced the mean number and volume of established lesions with corresponding diminished cell proliferation, reduced vascular density, and increased apoptosis within the lesions [379]. The ability of dietary resveratrol to inhibit angiogenesis and inflammation in endometriosis was demonstrated in a study on 24 female Wistar albino rats medicated for 21 days. After the treatment, significant reductions in the mean areas of the endometriotic implants and mean VEGF-staining scores of the endometriotic implants were confirmed. Moreover, the plasma fluid and serum levels of VEGF and MCP-1 were also significantly lower in the resveratrol-fed group [380]. The effect of polydatin (PLD, which is a natural potent stilbenoid polyphenol that is a natural precursor of resveratrol), which was orally administered in comicronized form with palmitoylethanolamide (PEA, which is an endogenous fatty acid amide possessing anti-inflammatory activity, but unlike resveratrol, having no free radical scavenging activity), was examined in an autologous rat model of surgically induced endometriosis. After 28 days of micronized (PEA/PLD) treatment at 10 mg/kg/day, the rats (*n* = 10) displayed a smaller cyst diameter, with an improved fibrosis score and decreased mast cell number. The combined use of PEA and PLD resulted in decreased angiogenesis (vascular endothelial growth factor), nerve growth factor, intercellular adhesion molecule, matrix metalloproteinase 9 expression, and lymphocyte accumulation. Furthermore, an anti-inflammatory effect was documented, as markers of inflammation were reduced, such as peroxynitrite formation, (poly-ADP)ribose polymerase activation, IκBα phosphorylation, and nuclear factor-κB translocation in the nucleus [381].

**Quercetin**, one of the major naturally occurring nontoxic flavonoids, e.g., in fruits (grapes and peaches) and vegetables (onions and garlic), has caught the attention of those seeking endometriosis treatments because of its antioxidant, anti-inflammatory, anti-apoptotic, and neuroprotective properties, demonstrated in a safe dose range [382,383]. In a study on a rat model of endometriosis (*n* = 15), the beneficial effects of quercetin administered for 30 days at a dose of 15 mg/kg/day, especially in combination with 200 mg/kg/day of metformin, were demonstrated [384]. The use of such a treatment resulted in limiting the growth of endometriosis foci and stabilizing the level of gene expression for mTOR and autophagy markers in ectopic endometrium. Clinical trials with an adequate number of patients and a solution to the problem of low bioavailability of quercetin are needed [383,385].

Injections of the highest doses of **genistein** (50 μg/g and 16.6 μg/g of the body weight) sustained intestinal mesentery implants of uterine (endometriotic) tissue in rats, whereas dietary genistein (250 or 1000 mg/kg) and a lower dose (5.0 μg/g of the body weight) of this phytoestrogen did not support the implants. These results, which were obtained after 3 weeks of daily injections or exposure to dietary genistein, may indicate that ER modulation and genistein bioavailability play a critical role in the maintenance of endometriotic implants [386]. In another previous study on female Wistar albino rats, in which endometriotic implants were induced by transplanting autologous uterine tissue to ectopic sites on the peritoneum, the results in the study group (*n* = 10) subjected to the oral administration of genistein at 500 mg/kg per day exhibited a statistically significant regression of endometriosis. After 3 weeks, a decrease in the surface area of the endometriotic implants was confirmed during histopathologic examinations with morphometry [387].

In a mouse model of endometriosis established by transplanting donor-mouse uterine fragments into recipient mice, the administration of a diet containing a mixture of principal isoflavonoids of soy (**daidzein + genistein + glycitein**) significantly decreased the number, weight, and Ki-67 proliferative activity of endometriosis-like lesions. According to the results of a parallel in vitro study on the effect of the combined administration of daizein, genistein, and glycitein on stromal cells isolated from ovarian endometrioma, the indicated anti-endometriotic effects may be related to the reduced expression of IL-6, IL-8, COX-2, and aromatase, as well as reduced aromatase activity, serum-glucocorticoid-regulated kinase levels, and PGE2 levels [388].

**Puerarin**, which is a hydroxyisoflavone glycoside originally isolated from Pueraria lobata (Willd.), is an isoflavone substituted by hydroxy groups at positions 7 and 4′, as well as a beta-D-glucopyranosyl residue at position 8 via a C-glycosidic linkage. The ability of this phytoestrogen to treat endometriosis was examined in female Sprague Dawley rats with endometriotic implants during 4 weeks of administration via oral gavage at doses of 600, 200, or 60 mg/kg per day. The endometriotic tissue weight and serum estrogen levels were significantly lower in the high-, medium-, and low-dose puerarin treatment groups than in the control group. Moreover, even low-dose puerarin inhibited aromatase cytochrome P450 (P450AROM) expression and reduced estrogen levels in endometriotic tissue. Furthermore, three doses of puerarin had no adverse effects on the liver, kidney, and ovary, whereas high-dose puerarin administration caused thinner bone trabeculae with distortion and breakage [389].

**Xanthohumol** is a prenylated flavonoid isolated from hops, and its effectiveness in the treatment of endometriosis was tested in BALB/C mice with surgically induced peritoneal and mesenteric endometriosis by uterine tissue transplantation into the abdominal cavity. After 28 days of daily treatment with 100 µM xanthohumol (*n* = 8) via drinking water, a marked reduction in the diameter of endometriotic lesions was observed (regardless of their location within the peritoneal cavity) compared with the control. This effect was accompanied by a reduced level of phosphoinositide 3-kinase (PI3K) protein. A significantly lower microvessel density documented within the xanthohumol-treated lesions indicates an inhibitory effect of this flavonoid on angiogenesis. Moreover, additional analyses demonstrated that treatment with xanthohumol did not affect the histomorphology, proliferation, or vascularization of the uterine horns and ovaries. The lack of serious side effects in the reproductive organs may be an advantage when considering the treatment of endometriosis [390].

The beneficial effects of **silymarin**, which is a compound with potent phytoestrogenic, proapoptotic, and antioxidative properties, were confirmed in a prospective study on rats with experimentally induced endometriosis (*n* = 12). After 28 days of oral silymarin administration (50 mg/kg per day), a significant decrease in the establishment and size of endometriotic lesions was noted with decreased mRNA levels of glial cell-derived neurotrophic factor (GDNF) and its essential receptor component, GFRα1, as well as the proto-oncogenes B-cell lymphoma 6 (Bcl-6b) and Bcl-2. The number of GDNF-, GFRα1-, Bcl-6b-, and Bcl-2-positive cell distribution/mm^2^ was remarkably diminished within endometriotic foci in the silymarin-treated group vs. the control. Moreover, silymarin promoted the apoptosis pathway by enhancing extracellular regulator kinase (ERK1/2) expression and by suppressing Bcl-2 expression. The authors of the study concluded that silymarin downregulates the angiogenesis ratio, accelerates apoptosis, and, consequently, induces severe fibrosis in endometriotic-like lesions [391].

A significant regression of surgically induced endometriotic foci in Wistar albino rats was observed after oral administration of the terpene **nerolidol** (trans-nerolidol) and the flavone glycoside **hesperidin**. Both PEs are potent antioxidants. In addition to a reduction in the average volume of the lesions in rats treated with hesperidin and nerolidol, malondialdehyde levels (the marker of oxidative stress) were significantly reduced in the nerolidol-treated group, and glutathione levels and superoxide dismutase activity (the first-line defense antioxidants) were significantly elevated in the endometriotic foci of both the hesperidin- and nerolidol-treated groups compared with the control endometriosis group [392].

The action of **isoliquiritigenin**, which is a natural flavonoid isolated from the root of licorice (*Glycyrrhiza uralensis*) and shallot (*Allium cepa*) with documented antioxidant, anti-inflammatory, antiproliferation, and antitumor activities, was examined in female BALB/C mice that were surgically induced to have endometriosis by transplanting uterine tissue into the abdominal cavity. Four weeks of oral administration of isoliquiritigenin reduced the volume and weight of endometriotic lesions, decreased serum and lesion inflammatory cytokines, induced apoptosis of the lesions, and inhibited the epithelial–mesenchymal transition (EMT) [393]. The latter effect should be emphasized because EMT, which is a process in which epithelial cells lose polarized organization of the cytoskeleton and cell-to-cell contacts, thus acquiring the high motility of mesenchymal cells, seems to be a prerequisite for the original establishment of endometriotic lesions [394].

**Naringenin** is a plant-derived flavonoid with anti-proliferative, anti-inflammatory, and anti-angiogenic properties in chronic and metabolic diseases. The therapeutic potential of orally administered naringenin in endometriosis was evaluated in a rat model of the disease. The endometrial lesion volumes, weight, serum TNF-α level, and histopathologic scores were significantly reduced in the naringenin-treated group compared to the endometriotic control group. Accordingly, naringenin ameliorated the expression of various proteins involved in the development and progression of endometriotic cells, such as p21-activated kinase 1 (PAK1), transforming growth factor β-activated kinase 1 (TAK1), VAGF, and proliferating cell nuclear antigen (PCNA). Moreover, in an in vitro study, naringenin caused a dose-dependent loss of mitochondrial membrane potential, induced apoptosis, and inhibited the proliferation/invasiveness of endometriotic cells with a corresponding downregulation of matrix metalloproteinase-2 (MMP-2) and 9 (MMP-9). The induction of reactive oxygen species (ROS)-mediated apoptosis was demonstrated by analyzing the effect of naringenin on the factor erythroid 2-related factor 2 (Nrf2)/heme oxygenase-1 (HO-1)/NADPH-quinone oxidoreductase-1 (NQO1)/Kelch-like ECH associated protein 1 (KEAP1) axis. This axis is a fundamental signaling cascade that controls multiple cytoprotective responses via the induction of a complex transcriptional program that finally provides endometriotic cells with increased resistance to oxidative, metabolic, and therapeutic stress. Naringenin significantly inhibited this transmission by modulating the expression of Nrf2 and its downstream effector molecules [395].

Similar results regarding the effect on the course of endometriosis were obtained by using oral flavonoid phytoestrogen-rich plant extracts in animal models, including surgically induced endometriosis in mice (female BALB/C) and female Sprague Dawley or Wistar albino rats. Significant regressions of the endometriotic foci were demonstrated after 3–4 weeks of administration of the extracts prepared from *U. dioica* L. (leaves and roots), the aerial parts of *Achillea millefolium* L., *Achillea biebersteinii Afan*, and *Achillea cretics* L.; *Rosmarinus officinalis* (leaves); *Scutellaria baicalensis* (root); *Anthemis austriaca* Jacq. (flowers); and *Melilotus officinalis* (L.) Pall. (aerial parts) [396,397,398,399,400,401]. The antiproliferative effects on the cells of the ectopic endometrium were accompanied by anti-inflammatory effects, which were manifested by decreased proinflammatory cytokine levels in the peritoneal fluid, including TNF-α, VEGF, and IL-6 levels [396,397,400,401].

### 3.3. PE Oral Intake and the Course of Endometriosis—The Results Obtained in Human Studies

The studies on the effect of orally administered PEs in women on the risk of onset and the course of endometriosis are summarized in Table 2. For this purpose, the electronic databases PubMed, EMBASE, and MEDLINE were searched until April 2023. The number of studies including randomized controlled trials that were identified in this manner was surprisingly few, especially considering the numerous studies that have been conducted on animals. In contrast, the number of comprehensive reviews oriented toward the role of diet in human health (particularly in endometriosis) is striking [169,171,402,403,404,405,406].

The results of most of the ten studies cited in Table 2 demonstrated some beneficial effects of a PE-rich diet or a diet supplemented with various doses of particular PE/PEs (i.e., resveratrol, daidzein + genistein, or quercetin + curcumin + parthenium) in endometriosis. This scenario applies to both the risk of disease onset and its course [407,408,409,410,411,412,413,414,415,416]. For example, the results of 15 years of two case-control studies (*n* = 504) have shown that a PE-rich diet decreases the risk of endometriosis compared to a low-PE diet [411]. The same effect was shown regarding a decreased risk of endometriosis for isoflavones, lignans, and coumestrol, although this was observed in a much smaller 12-month case-control study on dietary data (*n* = 78) [415]. Moreover, increased urinary levels of soy isoflavones (daidzein and denistein) were inversely associated with both the risk of advanced endometriosis (stages III–IV) and the severity of endometriosis. Importantly, the authors of this case-control study (*n* = 79) within 24 months of the recruiting period noted that the ERβ gene *ESR2* RsaI polymorphism significantly modified these effects of genistein for advanced endometriosis [414]. In addition, resveratrol appeared in three small studies [407,408,409] (Table 2). In a retrospective study (*n* = 26), 82% of patients reported complete resolution of dysmenorrhea and pelvic pain after 2 months of treatment with 30 mg oral administration of resveratrol per day. This effect was accompanied by reduced expression of both COX-2 and aromatase in the eutopic endometrium [408]. In a placebo-controlled, randomized, double-blind clinical trial (*n* = 17), the dose of resveratrol was considerably higher (400 mg daily for 12–14 weeks). Resveratrol has been shown to have anti-inflammatory effects by affecting the metalloproteinases MMP-2 and MMP-9. Furthermore, the decrease in the mRNA and protein levels of both MMP-2 and MMP-9 was significant. As a consequence, decreased concentrations of MMP-2 and MMP-9 in the serum and endometrial fluid were confirmed [407].

In contrast, the ineffectiveness of resveratrol in endometriosis pain relief was demonstrated in a short randomized clinical trial. After 42 days of treatment with a dose of 40 mg, resveratrol was not superior to placebo (no difference in median pain scores observed between the groups) [409].

Research by another team provided new information on the role of dietary factors in the development of endometriosis [413]. The results of this 60-month, population-based case-control study involving 300 patients in the study group demonstrated that an increased risk of endometriosis is positively correlated with β-carotene consumption and servings/d of fruit. Unlike fruits served twice a day, vegetable intake was not associated with endometriosis risk. These findings were not consistent with the reduced risk of endometriosis associated with the consumption of both green vegetables and fresh fruit reported by Parazzini et al. [411]. The authors realized that their results require confirmation or even validation because the risk of misinterpretation of the data is high due to the number of included variables. A study by Parazzini demonstrated partial confirmation in a paper published by Harris et al. [416], in which a reduced risk of endometriosis was reported when consuming significant amounts of fruit (especially citrus) in the diet (but not for vegetables). Moreover, due to the much larger size of the cohort, these results may be more convincing. To date, no similar prospective studies with PEs have been conducted on humans. A more detailed understanding of the impact of dietary PEs, including mycotoxins and dietary patterns, on the risk of endometriosis is urgently needed [48,49,148,166,167]. Furthermore, it may help to develop population-based strategies to prevent this chronic disease with a high impact of environmental (possibly dietary) triggers. A key question for those individuals already suffering from endometriosis may be whether a diet rich in phytoestrogens is truly beneficial [169].

The inconclusive or (less commonly) opposing results are understandable with so few human studies and their varied methodologies (e.g., differences in the amount and content of PEs in the diet, different duration of the study and/or size of the sample, imprecise inclusion/exclusion criteria, and interpretation of results of measured outcomes in inconsistent manners) [417].

## 4. Concluding Remarks

There is no doubt that PEs may have antiestrogenic activities typical of substances from the group of endocrine disruptors [46,137]. If absolute or relative excess estrogen plays a key role in endometriosis, the antiestrogenic effect of PEs may provide therapeutic benefits. In vitro studies in cell cultures and experiments on animals using various types of endometriosis models are widely represented, as well as in the latest literature. In addition to zearalenone and other mycotoxins, the most commonly studied phytoestrogens in this manner have been resveratrol, curcumin, soy isoflavones (daidzein and genistein), lignans, coumestrol, quercetin, epigallocatechin-3-gallate, parthenolide (parthenium), puerarin, ginsenosides (steroid-like saponins), xanthohumol, and cannabinoids (apigenin) [418,419]. The characteristics of the detected pleiotropic effects of these xenoestrogens are related to a variety of known signaling effectors, including ERs, GPER, COX-2, IL-1, IL-6, TNFα, VEGF, ROS, MMPs, NF-κB, and apoptosis-related proteins (e.g., Bax, Bcl-2, Caspase-3, Caspase-9, p53, and β-actin) [419,420]. In most such experimental models, solid evidence about the anti-inflammatory, proapoptotic, antioxidant, and immunomodulatory functions of phenolic compounds (e.g., flavonoids and phenolic acids) have translated into an effect of inhibiting the development of endometriosis via the modulation of estrogen activity [406].

However, one should be fully aware that these promising results are based on in vitro and animal models of endometriosis. As the summary in Table 2 shows, there are virtually no randomized controlled trials in this area. Thus, to achieve conclusive results regarding the potential benefits of oral (dietary) phytoestrogens, properly designed clinical trials are essential. When developing schemes for such studies, several (often underestimated) issues should also be considered (Figure 6).

Research has demonstrated a connection between diet (which is the most important environmental epigenetic factor) and the incidence of estrogen-dependent diseases (e.g., breast and endometrial cancer) [421,422]. With regard to endometriosis, it has been established that fish oil capsules in combination with vitamin B_12_ can be helpful in relieving endometriosis symptoms (especially dysmenorrhea), whereas alcohol and increased consumption of red meat and trans fats are associated with an exacerbation of the disease [184]. However, the existing information about the effect of dietary phytoestrogens on endometriosis in humans is still incomplete, inconclusive, or ambiguous. This is an important topic to explore because endometriosis affects up to 10% of women, and diet is a modifiable risk factor for this chronic disease, both in terms of onset and management. The analysis of the results cannot ignore the possible conflict of interest caused by the source of funding for such research [169,423].

An evaluation of the effects of PEs should consider differences between populations and different nutritional patterns. For example, the prevalence of endometriosis appears to be higher in Asian women (e.g., Filipino, Indian, Japanese, and Korean women) than in Caucasian women and African Americans. Although the utilization of health care may account for some of the observed differences, the incidence of endometriosis is estimated at 5–10% in Western populations, compared with 15–18% in Asian female populations [424,425,426]. This result is puzzling because a PE-rich diet is the basis in most Asian countries. For example, in the context of the beneficial effects of PEs on endometriosis, it is surprising that India (the country with the highest percentage of vegans/vegetarians) has a high incidence of endometriosis [425,427,428]. It is suggested that a genetic polymorphism predisposing to endometriosis and/or environmental pollution with other EDCs (e.g., pesticides used in plant cultivation) may be of importance in this scenario [425].

It should be noted that the family of PEs includes compounds of different strengths and specificities of action on receptors and signaling pathways. These actions are sometimes contradictory. For example, resveratrol inhibits aromatase, thus lowering the concentration of estrogens, whereas genistein has the opposite effect on aromatase activity in human endometrial stromal cells [251,429].

In accordance with the principle known from toxicology that “the dose defines the poison”, to distinguish genotoxic from beneficial effects, we need to know the bioavailability of PEs in a given person with a specific health condition [430]. Correspondingly, the bioavailability, bioactivity, and health effects of dietary PEs are strongly determined by the intestinal bacteria of each individual [431]. The gut microbiota regulates estrogenic activity in the body through the secretion of β-glucuronidase, which is an enzyme that deconjugates estrogens into their active forms. Notably, the “carrier” of PEs is a diet rich in fiber. Dietary fiber has been shown to affect the absorption, reabsorption, and excretion of estrogens and PEs by influencing the ß-glucosidase and ß-glucuronidase activities of the intestinal microflora [432]. The swelling of the fiber leads to the dilution of the intestinal bacterial flora and hydrophobic bonding, particularly of nonconjugated compounds, which contributes to the reduced absorption of PEs and the reabsorption of endogenous estrogens. Therefore, the antiestrogenic effects of PEs may be variable and secondary to dietary fiber content [433]. High dietary fiber intake may lead to the partial disruption of enterohepatic circulation of estrogens within the estrogen–gut microbiome axis [359,434]. Accordingly, vegetarians generally have higher fecal weights than omnivores and lower fecal bacterial ß-glucuronidase activity [435]. There is evidence that dysbiotic gut microbiota and dysfunctional estrobolome (which represents the aggregate of all of the enteric bacteria capable of metabolizing estrogen) are associated with multiple gynecologic conditions, with mounting data supporting an association between the intestinal bacteria and endometriosis and infertility [370]. In such cases (including endometriosis), bacteria-derived/induced metabolites may interact with the cells of the immune system and nervous system, thus modulating the actions of these systems [177,184].

In summary, due to the lack of relevant studies on humans, it is impossible to unequivocally determine the benefits or adverse effects of using a diet rich in PEs in relation to the risk of endometriosis or its course [169,436,437,438,439].

## Figures and Tables

**Figure 1 ijms-24-12195-f001:**
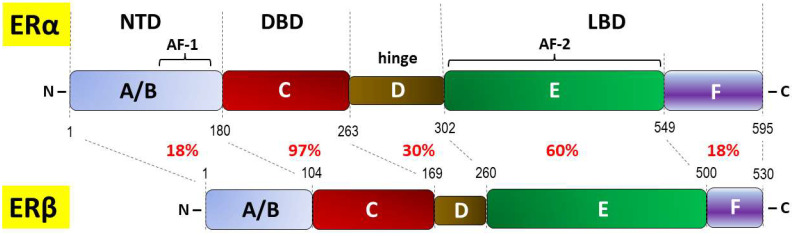
Structural and functional domains of estrogen nuclear receptors: ERα vs. ERβ. Both receptors have six crucial different domains marked with the letters A to F: N-terminal domain (NTD, A/B domain containing AF-1 domain), DNA-binding domain (DBD, C domain), D domain (hinge), and ligand-binding domain (LBD, E/F domain containing AF-2 domain) at the C-terminal region. Homology between ERα and ERβ, understood as a percentage (%) of amino acid identity within the respective domains, is also shown [61,62,63].

**Figure 2 ijms-24-12195-f002:**
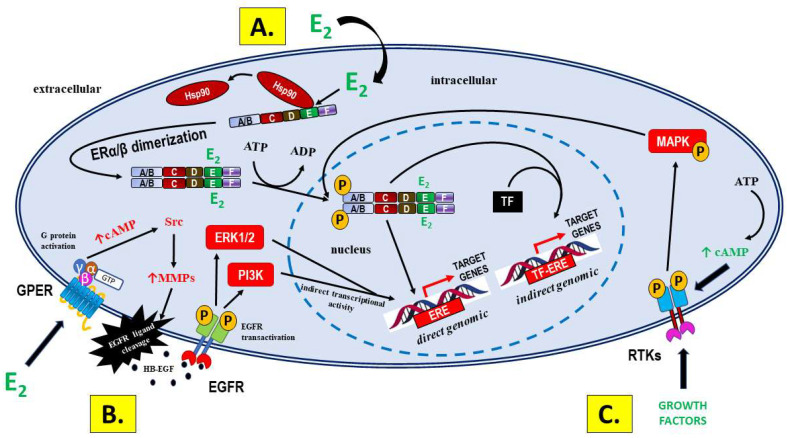
Mechanisms of estrogen signaling. (**A**) “Classic” ligand-dependent genomic signaling via nuclear receptors ERα/ ERβ: estrogen binding to the ligand-binding domain (LBD) unblocks the receptor with the release of “inhibitory” heat shock protein 90 (Hsp90) and subsequent ER dimerization; ER is then translocated from the cytoplasm to the nucleus and activated by phosphorylation (P); in the nucleus, ER acts as ligand-activated transcription factor and exerts both direct and indirect genomic activity through binding of DNA-binding domain (DBD) to the estrogen response element (ERE) on the target gene and via interaction with transcription factor (TF), which enables binding of DBD to the ERE as TF-ERE, respectively [61,62,63]. (**B**) Indirect, rapid, non-genomic signaling via membrane-associated G protein-coupled estrogen receptor (GPER) and transactivation of the epidermal growth factor receptor (EGFR): GPER stimulation activates non-receptor tyrosine kinase (proto-oncogene tyrosine-protein kinase Src), which increases the concentration of matrix metalloproteinases (MMPs), resulting in EGFR ligand cleavage; indirect transcriptional activity may occur because released heparin-binding EGF-like growth factor (HB-EGF) produces downstream activation of mitogen-activated protein serine/threonine kinases (ERK1 and ERK2) and phosphatidylinositol-3-kinase (PI3K) pathways [64,65]. (**C**) The ligand-independent pathway on the example of growth factors signaling through receptor tyrosine kinases (RTKs); growth-factor-receptor-specific ligands bind to the extracellular regions of RTKs and interact with cAMP to activate RTKs (activate the receptor tyrosine kinases) and a mitogen-activated protein kinase (MAPK); MAPK can then phosphorylate and activate ERα/ ERβ either independent of E_2_ or in synergy with E_2_ [66,67,68].

**Figure 3 ijms-24-12195-f003:**
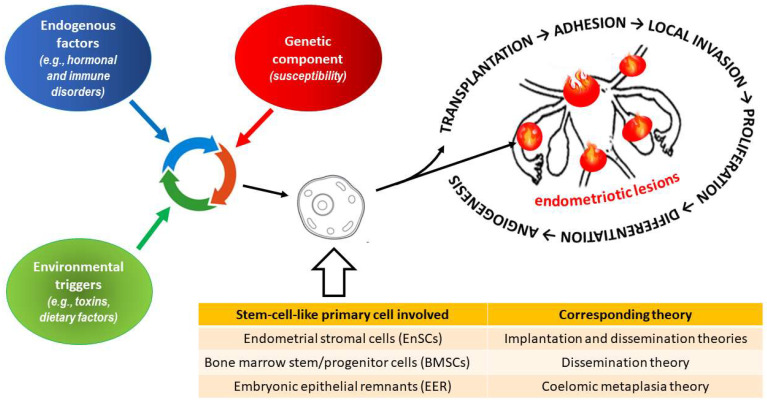
Theories on etiopathogenesis of endometriosis. It is assumed that the development of endometriotic foci is a consequence of the dissemination and transplantation of the cells with clonogenic activity or gradual transformation of embryonic duct remnants. The specific system of interactions between genetic, endogenous, and environmental factors determines the occurrence of the disease [206,207,213,214,215,218,219].

**Figure 4 ijms-24-12195-f004:**
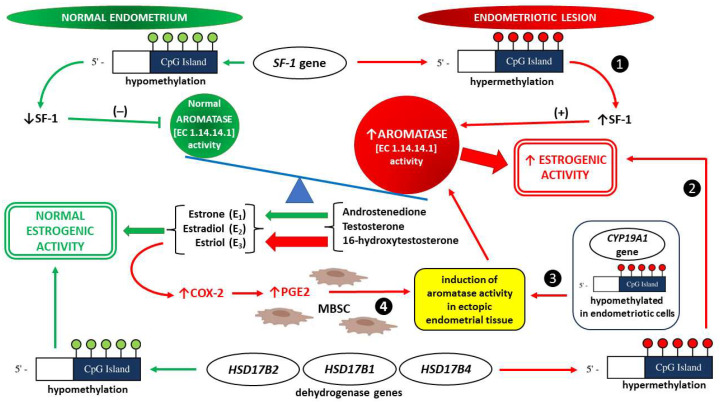
Modulation of aromatase by epigenetic factors and estrogenic activity: normal endometrium (pathways in green) vs. endometriotic lesion (pathways in red). Estrogenic hyperactivity in endometriosis is caused by: ❶—hypermethylation of CpG island in the transcriptor factor steroidogenic factor 1 (SF-1) gene; ❷—deficient 17β-hydroxysteroid dehydrogenases expression due to hypermethylation of the respective genes (*HSD17B2*, *HSD17B1*, *HSD17B4*); ❸—aromatase [EC 1.14.14.1] gene (*CYP19A1*) activation due to CpG islands hypomethylation; ❹—positive feedback: estrogens → cyclooxygenase 2 (COX-2) → prostaglandin E_2_ (PGE2) → aromatase activity in menstrual blood stem cells (MBSC).

**Figure 5 ijms-24-12195-f005:**
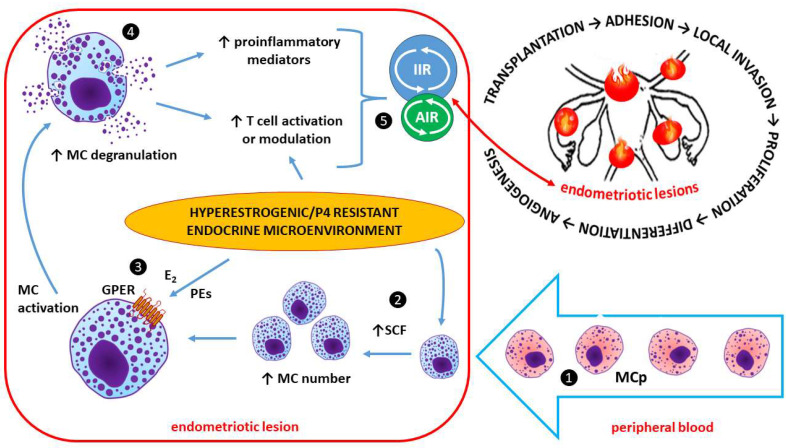
Estrogen-dependent innate and adaptive immune responses related to mast cells (MCs) in endometriotic foci. ❶—MCs develop from mast cell progenitors (MCp), which migrate to the peripheral tissues, including endometriotic foci, via blood circulation. ❷—When stimulated by estrogens, e.g., estradiol (E_2_) or phytoestrogens (PEs), endometrial-like tissue within the endometriotic lesion produces increased amounts of stem cell factor (SCF), a potent growth factor critical for MC expansion, differentiation, and survival for tissue resident MCs. ❸—Among the estrogen receptors, G protein-coupled receptor (GPER) is responsible for the various running-fast nongenomic effects of estrogens, including activation and degranulation of MCs. ❹—MCs may release a diverse spectrum of mediators that contribute to inflammation, chronic pelvic pain, and local angiogenesis, resulting in the disease progression. ❺—The release of granular and secreted mediators from MCs modulate innate immune response (IIR). An altered adaptive immune response (AIR) is observed, mainly due to MC-dependent enhancement of T-cell activation. Moreover, estrogens have been shown to modulate all subsets of T cells that include CD4+ (Th1, Th2, Th17, and Tregs) and CD8+ cells (cytotoxic T lymphocytes, or CTLs) [297,312,350].

**Figure 6 ijms-24-12195-f006:**
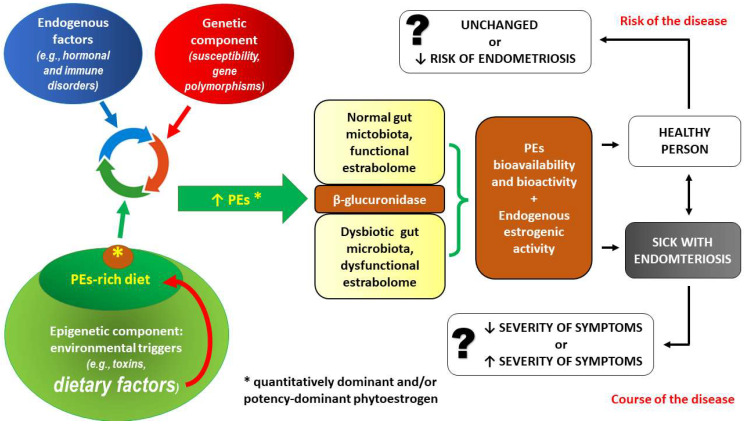
Key issues and factors potentially affecting the beneficial, neutral, or adverse effects of phytoestrogens (PEs) in endometriosis. Due to the lack of adequate randomized controlled clinical trials on a representative sample of women, a difficult unequivocal answer regarding the recommendation of PEs in endometriosis remains unclear. Due to expected clinically significant heterogeneity in response to PEs, some patients will experience more or less benefit from the treatment of endometriosis than the averages, while other patients may experience an exacerbation of the disease.

**Table 1 ijms-24-12195-t001:** Phytoestrogens (PEs)—an overview of the family of naturally occurring polycyclic phenols.

Class of PEs	Subgroups	Basic Chemical Structure	Examples
Flavonoids	Flavanols	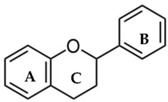 Flavan	Myricetin, kaempferol, fisetin, rhamnazin
	Flavones	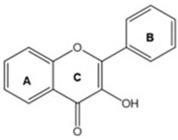 Flavonol	Apigenin, luteolin, tangeritin
	Anthocyanidins	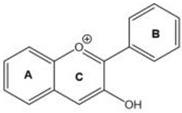 Anthocyanidin	Cyanidin, malvidin
	Isoflavonoids	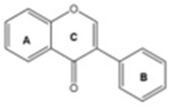 Isoflavone	Isoflavones: genistein, daidzein, glycitein
	Neoflavonoids	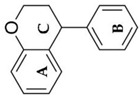 Neoflavan	Neoflavan, dalbergin, nivetin
Stilbens	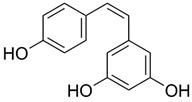 *cis*-resveratrol	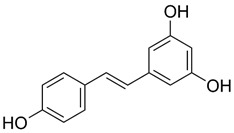 *trans*-resveratrol	Resveratrol, pterostilbene, rhapontigenin
Entero-lignans	Dibenzylbutyrolactones (type 1)	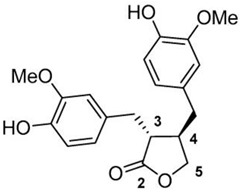 (-)-Matairesinol	Matairesinol, arctigenin
	Dibenzylbutanediols (type 2)	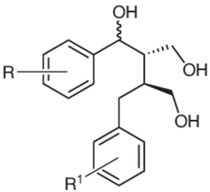 Dibenzylbutanediol	1,4-butanediol terephthalate dibenzyl ester, 2,3-butanediol, 1,3,-butanediol
	Dibenzyltetrahydrofurans (type 3)	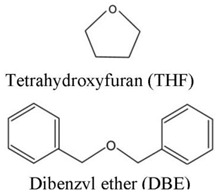	Tetrahydrofuran, dibenzyl tetrahydrofuran
Coumestans	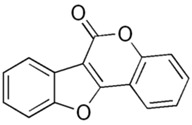 Coumestan		Coumestrol, wedelolactone, psoralidin, glycyrol
Pterocarpans	Medicarpin 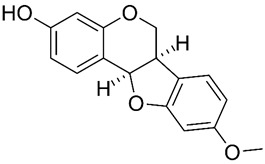		Medicarpin, bitucarpin A, bitucarpin B, phaseolin
Mycotoxins	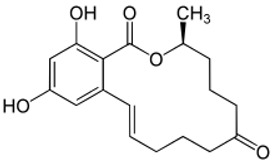 Zearalenone (ZEN)		Zearalenone, aflatoxins, ergot alkaloids (ergotamine), citrinin, fumonisins

**Table 2 ijms-24-12195-t002:** Human studies on the effect of phytoestrogens (PEs) administered orally on endometriosis [169]. Two studies were included wherein PEs were analyzed together with other nutrients in the diet [407,408]. In such studies, only endometriosis-related outcomes are summarized.

Authors	Year	Type of the Study	Compound(s), Duration	Sample Size (*n*)	Age Range (Years, Mean)	Control (*n*)	Main Results (*p* < 0.05)	LoE *
Kodarahmian et al. [409]	2019	Placebo-controlled, randomized, double-blind clinical trial	Resveratrol 400 mg; 12–14 weeks	17	18–37 (30.19 ± 2.4)	17 (placebo)	- ↓ level of mRNA and protein of both MMP-2 and MMP-9; - ↓ concentration of MMP-2 and MMP-9 in the serum and the endometrial fluid.	II
Maia Jr et al. [410]	2012	Retrospective study	Resveratrol 30 mg; 2–6 months	26 using OC	24–40 (31 ± 4.0)	16 using OC	- ↓ pain (82% of patients reporting complete resolution of dysmenorrhea and pelvic pain after 2 months); - ↓ expressions of both COX-2 and aromatase in eutopic endometrium.	II
Mendes da Silva et al. [411]	2017	Randomized clinical trial	Resveratrol 40 mg; 42 days	22 using MOC	20–50 (35.4 ± 7.1)	22 using MOC (placebo)	- No difference in median pain scores between the groups; - Resveratrol was not superior to placebo for treatment of pain in endometriosis.	III
Nagata et al. [412]	2001	Prospective cohort study	Soy isoflavones: daidzein and genistein; 6 years	1172	35–54 (42.9 ± 4.4)	N/A	- ↓ risk of premenopausal hysterectomy: RR (95% CI) 0.35 (0.13–0.97).	II
Parazzini et al. [407]	2004	Two case-control studies	PE-rich vs. low-PE diet; 15-year data	504	Cases: 20–65 (33 ± 3.3) Controls: 20–61 (34 ± 2.9)	504	- ↓ risk of endometriosis for PE-rich diet (OR = 0.3 for the highest tertile of intake for green vegetables, and OR = 0.6 for fresh fruit).	III
Signorile et al. [413]	2018	Prospective, placebo-controlled, cohort study	Dietary supplement containing quercetin (200 mg), curcumin (turmeric curcumin 20 mg), parthenium (19.5 mg); 3 months	34	NP	30 (placebo)	- ↓ symptoms in endometriosis: dysmenorrhea and chronic pelvic pain (both from 62% to 18%), dyspareunia (from 30% to 15%); - ↓ serum levels of PGE2 and CA-125.	III
Trabert B et al. [408]	2011	Population-based case control study	Overall intake of fruits (excluding fruit juice), vegetables, dairy, whole grains, legumes, red meat, poultry, fatty fish, nonfatty fish and seafood; 60 months	284	Cases: 18–49 (NP) Controls: 18–49 (NP)	660 (randomly selected, without a history of endometriosis)	- ↑ risk of endometriosis positively correlated with β-carotene consumption and servings/d of fruit, whereas vegetable intake was not associated with endometriosis risk.	I
Tsuchiya et al. [414]	2007	Case-control study	Urinary levels of soy isoflavones: daidzein and genistein; 24 months of recruiting period	79 (stages I–II: 31) (stages III–IV: 48)	20–45 (stages I–II: 32.3 ± 3.2) (stages III–IV: 32.6 ± 3.7)	59	- ↑ urinary level of isoflavones was inversely associated with both the risk of advanced endometriosis (stage III–IV) and severity of endometriosis; - For advanced endometriosis, ER2 gene *RsaI* polymorphism significantly modifies the effects of genistein.	III
Youseflu et al. [415]	2020	Case-control study on dietary data	Isoflavones, lignans, and coumestrol; 12 months	78	15–45 (31.01 ± 6.56)	78	- ↓ risk of endometriosis for isoflavones, lignans, and coumestrol.	III
Harris et al. [416]	2018	Prospective cohort study	Intake of fruits and vegetables; 22-year follow-up period	70,835	25–42 (NP)	N/A	- ↓ risk of endometriosis for higher fruit consumption, especially for citrus fruits; - ↓ risk of endometriosis was positively correlated with β-Cryptoxanthin intake; - No association between total vegetable intake and endometriosis risk.	

* Level of evidence (LoE) for clinical studies, according to the Levels of Evidence for Primary Research Question adopted by the North American Spine Society January 2005; CI—confidence interval; COX-2—cyclooxygenase-2; ER2—estrogen receptor-2; MMP-2, MMP-9—matrix metalloproteinases 2 and 9, respectively; MOC—monophasic oral contraceptive; N/A—not applicable; NP—not provided; OC—oral contraceptive; OR—odds ratio; RR—rate ratio.

## Data Availability

No new data were created. Instead, the data are quoted from the available cited literature.

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
