# Peer review of "Insight into the Potential Mechanisms of Endocrine Disruption by Dietary Phytoestrogens in the Context of the Etiopathogenesis of Endometriosis"

_ijms, 2023, doi:10.3390/ijms241512195_

Round 1

Reviewer 1 Report

Endometriosis is a disease that affects fertility potential. In addition, the role of cellular senescence in female reproductive aging and the potential for senotherapeutic interventions (e.g., Secomandi et al., Human Reproduction Update, Vol.28, No.2, pp. 172–189, 2022) in women of advanced reproductive age are urgent.

INFLAMMAGING is the aging mechanism. Ex. quercetin is considered one of the most studied flavonoids. Quercetin plays a role in age-related diseases. Dietary compounds such as quercetin should be discussed.

Reviewer 2 Report

Review report

   The manuscript entitled “Insight into the potential mechanisms of endocrine disruption by dietary phytoestrogens in the context of the etiopathogenesis of endometriosis” Prepared by Szukiewicz is well-designed and prepared. The discussed subject, endometriosis, is one of the common reproductive issues affect wide populations worldwide. In my opinion, the author has successfully covered this subject and highlighted the major causes of endometriosis along with its molecular pathogenesis. The used language was scientific and simple and easy to catch. Cited references are adequately used and approximately more than 50% of them were about 5 years old or newer which is one of the strengths of this work. However, some minor notes were raised and need to be addressed to improve the quality of this manuscript.

Minor concerns:

1-     As the manuscript is structured by one author, there is no need to put indicators as asterisk or number to refer to affiliation or assign as corresponding author.

2-     In section 3. “Dietary PEs and endometriosis”; 3.1 microbiome, It is better to add some info about the bacteria that metabolize the steroids, particularly the estrogen. And adding more about the bacterial gus gene, which regulates the glucuronidase enzyme production and activity.

3-     Line 1037 (L1037): correct periteum to peritoneum.

4-     The manuscript lacks to the role of myeloid-derived suppressor cells (MDSCs) in the pathogenesis of endometriosis.

The quality of writing is excellent.
